

# Multiple-scattering effects on single-wavelength lidar sounding of multi-layered clouds

Valery Shcherbakov[1], Frédéric Szczap[1], Guillaume Mioche[1,2], Céline Cornet[3]

[1]Université Clermont Auvergne, CNRS, UMR 6016, Laboratoire de Météorologie Physique, 63178 Aubière, France
[2]Université Clermont Auvergne, Institut Universitaire de Technologie Clermont Auvergne – site de Montluçon, 03100 Montluçon, France
[3]Université Lille, CNRS, UMR 8518 - LOA - Laboratoire d'Optique Atmosphérique, F-59000 Lille, France

*Correspondence to*: Valery Shcherbakov (v.shcherbakov@opgc.univ-bpclermont.fr)

**Abstract.** We performed Monte Carlo simulations of single-wavelength lidar signals from multi-layered clouds with special
attention focused on multiple-scattering (MS) effect in regions of the cloud-free molecular atmosphere, i.e. between layers or outside a cloud system. Despite the fact that the strength of lidar signals from molecular atmosphere is much lower compared to the in-cloud intervals, studies of MS effects in such regions are of interest from scientific and practical points of view.

The MS effect on lidar signals is always decreasing with the increasing distance from the cloud far edge. The decreasing is the direct consequence of the fact that the forward peak of particles phase functions is much larger than the receiver field of view
(RFOV). Therefore, the photons scattered within the forward peak escape the sampling volume formed by the RFOV, i.e. the escape effect. We demonstrated that the escape effect is an inherent part of MS properties within the free atmosphere beyond the cloud far edge.

In the cases of the ground-based lidar, the MS contribution is lower than 5% within the regions of the cloud-free molecular atmosphere having the distance from the cloud far edge about 1 km or higher. In the cases of the space-borne lidar, the
decreasing rate of the MS contribution is so slow that the threshold of 5% can hardly be reached. In addition, the effect of non-uniform beam filling is extremely strong. Therefore, practitioners should employ with proper precautions lidar data from regions below the cloud base when treating data of a space-borne lidar.

In the case of two-layered cloud, the distance of 1 km is sufficiently large that the scattered photons emerging from the first layer do not affect signals from the second layer when we are dealing with the ground-based lidar. In contrast, signals from
the near edge of the second cloud layer are severely affected by the photons emerging from the first layer in the case of a space-borne lidar.

We evaluated the Eloranta model (EM) in extreme conditions and showed its good performance in the cases of ground-based and space-borne lidars. At the same time, we revealed the shortcoming that can affect practical applications of the EM. Namely, values of the key parameters, i.e. the ratios of phase functions in the backscatter direction for the n[th]-order-scattered photon
and a singly scattered photon depend not only on the particles phase function, but also on the distance from a lidar to the cloud and the receiver field of view. Those ratios vary within a quite large range and the MS contribution to lidar signals can be largely overestimated or underestimated if erroneous values of the ratios are assigned to the EM.



# 1 Introduction

It is well recognized that multiple scattering (MS) inevitably affects data of space-borne lidars (see e.g. Winker, 2003; Young and Vaughan, 2009; Shcherbakov et al., 2022). As for ground-based lidars, the MS relative contribution to lidar signals can exceed 20 % even in the case of cirrus clouds having the extinction coefficient $\varepsilon_p = 0.2$ km$^{-1}$, and can reach 200 % when $\varepsilon_p =$ 1.0 km$^{-1}$ (Shcherbakov et al., 2022). On the other hand, the MS effect is within measurement errors of ground-based and airborne lidars, i.e. the single-scattering (SS) approximation can be used, when the receiver field of view (RFOV) is quite narrow and/or the extinction coefficient is quite low (Shcherbakov et al., 2022).

Lidar signals are used as input data to retrieve profiles of characteristics of interest, e.g. the extinction coefficient, the backscatter coefficient, the lidar ratio (see e.g. Young and Vaughan, 2009). If the MS contribution cannot be neglected, a solution to the corresponding inverse problem has to take into account the MS effect in order to avoid biased retrievals. To put it differently, the correction for the effects of multiple scattering has to be applied (Young and Vaughan, 2009).

Generally, any algorithm to solve an inverse problem is based on a solution to the corresponding direct, i.e. forward problem (see e.g. Rodgers, 2000; Neto and Neto, 2013). It is obvious that retrieval quality is closely related to the accuracy of the direct modelling. To put it in relation to lidar sounding, the quality of the correction for MS effects directly depends on the accuracy of lidar-signals modelling in multiple-scattering conditions.

Expressed mathematically, any modelling in MS conditions has to be based on the radiative transfer equation and corresponding boundary conditions (see e.g. Marchuk et al., 2013). It is well known that there exist only a few cases when a solution to the radiative transfer equation can be obtained as an analytical expression. Thus, numerical methods, e.g. Monte Carlo (MC) simulations (Marchuk et al., 2013) or approximate models are largely applied to obtain solutions. Referring to lidar sounding, a good review of developed approximate methods and models can be found in (Bissonnette, 2005). Using some cases as examples, good performance of approximate models was underlined by their authors. At the same time, we believe that the accuracy level and the applicability bounds of approximate models still need to be rigorously evaluated.

It was suggested in the work by Weinman (1968) to approximate a phase function of cloud particles by a somewhat smoothly varying function of the scattering angle plus a narrow Gaussian peak for small angles. That idea was used to develop models that simulate lidar signals in multiple-scattering conditions and promising results were obtained (see e.g. Eloranta, 1998; Hogan, 2006; Hogan, 2008). For example, a good agreement with MC simulations of second-, third-, and fourth-order scattering was shown for the case of the extinction coefficient of 16.7 km$^{-1}$ (Eloranta, 1998). Moreover, the models (Eloranta, 1998; Hogan, 2006; Hogan, 2008) were employed by practitioners to account for MS effects while retrieving clouds' optical properties on the bases of experimental lidar data (see e.g. Nakoudi et al., 2021; Seifert et al., 2007; Delanoë and Hogan, 2008). On the other hand, results of MC simulations published in the literature (see e.g. Flesia and Starkov, 1996; Donovan, 2016; Reverdy et al., 2015; Szczap et al., 2021) evidenced the following. As it is expected, lidar signals from regions of the cloud-free molecular atmosphere, namely, between cloud layers and/or beyond the far edge of a cloud system, are affected by the





scattered light emerging from clouds. The effect on lidar signals of the emerging light has distinctive features that are not sufficiently addressed in the literature.

The main objective of this work is to perform Monte Carlo simulations of single-wavelength lidar signals from multi-layered clouds with special attention focused on peculiarities of MS effect in regions of the cloud-free molecular atmosphere, i.e. between layers or outside a cloud system. In our opinion, such study helps to obtain further insight into the problem of MS

effects. At the same time, it has practical applications because some of inverse-problem algorithms use lidar data taken from the range of the cloud-free atmosphere beyond the far edge of a cloud, for example, the two-way transmittance method (see e.g. Young and Vaughan, 2009; Giannakaki et al., 2007; and references therein).

The second objective of this work is to evaluate performance of an approximate model with the focus on the cloud-free regions. We have chosen for the evaluation the model developed by Eloranta (Eloranta, 1998) because multiple integrals are in its core

(see Eq. (11) in (Eloranta, 1998) and Eqs. (3-4) in (Eloranta, 2000)). Therefore, it is an easy matter to develop the corresponding code.

Throughout this work, the majority of results are shown and discussed in terms of relative contributions of multiple scattering (see definitions in Section 2.1), that is, multiple-to-single scattering lidar-return ratios (Bissonnette et al., 1995). At the same time, a reader should keep in mind that the ratios show not lidar signals but the relative contribution of MS with respect to

lidar signals obtained under the SS approximation. From the point of view of the inverse problem, namely the ratios are of importance because they show the level of retrieval errors if the SS approximation is applied in an inversion algorithm. If different models, which simulate the direct problem, are compared, the ratios evidence clearly the differences between the results (see e.g. Bissonnette et al., 1995). In our opinion, a comparison of lidar signals instead of ratios suffers from a severe disadvantage for the following reasons. Even when lidars signals are corrected for the offset and instrumental factors, they do

decrease exponentially because of the extinction. Consequently, the semilog plot is usually employed to show data, and even important differences are hardly noticeable in such figures. Moreover, when differences are seen, it is difficult for a reader to estimate their importance from the point of view of the inverse problem.

For brevity sake, the term "sampling volume" is used in this work in reference to the volume bounded by the RFOV of a lidar in the 3D space.

Section 2 addresses methodology of our Monte-Carlo simulations in details. Sections 3 and 4 show our simulation results for a ground-based and a space-borne lidars, respectively. Section 5 is devoted to conclusions. Appendix is focused on the Eloranta model and its input parameters in homogeneous-cloud conditions.

## 2 Methodology

### 2.1 Background

We use the following notations in this work. The function $S_1(h)$ characterizes lidar signals under the SS approximation (corrected for the offset, instrumental factors, and the two-way ozone transmittance):



$$S_1(h) = [\beta_p(h) + \beta_m(h)] \cdot T^2(h) = [\beta_p(h) + \beta_m(h)] \cdot T_m^2(h) \cdot T_p^2(h), \qquad (1)$$

where $h$ is the distance from the lidar; $\beta_p(h)$ and $\beta_m(h)$ represent the backscatter contributions from particles and from the atmospheric molecules; $T^2(h) = T_m^2(h) \cdot T_p^2(h)$ is the two-way transmittance from the lidar to the range $h$; $T_m^2(h)$ and $T_p^2(h)$

are the molecular and the particulate transmittances, respectively. $T_p^2(h) = 1$ if $h \leq h_b$, where $h_b$ is the distance to the cloud near edge; $T_p^2(h) = \exp[-2\tau_p(h_b, h)]$ when $h \geq h_b$, where $\tau_p(h_b, h) = \int_{h_b}^{h} \varepsilon_p(h')dh'$ is the cloud optical depth, $\varepsilon_p(h)$ is the extinction coefficient of particles.

The notation $S_k(h)$, $k = 2, 3, \ldots, n$ represents lidar signals (corrected for the offset, instrumental factors, and the two-way ozone transmittance) when all scattering events from first (single scattering) up to $k^{th}$ inclusive are taken into account; $(k+1)^{th}$

and higher orders of scattering are neglected. For example, $S_2(h)$ is the double scattering approximation (see e.g. Bissonnette, 2005). The notation $S_{MS}(h)$ means that all scattering events are taken into consideration. Of course, only $S_{MS}(h)$ can be obtained from real lidar measurements. $S_k(h)$ are useful data of simulations computed with the Monte Carlo method or an approximate model.

We are using below the following characteristics to compare the simulations results

$$R_{kto1}(h) = [S_k(h) - S_{k-1}(h)]/S_1(h), \qquad (2)$$

$$R_{MSto1}(h) = [S_{MS}(h) - S_1(h)]/S_1(h). \qquad (3)$$

The ratio $R_{kto1}(h)$ is the relative contribution of the $k^{th}$ order of scattering to a lidar signal. For example, $R_{4to1}(h) = [S_4(h) - S_3(h)]/S_1(h)$ provides the relative contribution of fourth order of scattering. The ratio $R_{MSto1}(h)$ is the relative contribution of multiple scattering to a lidar signal, i.e., all orders of scattering were taken into consideration. It is

evident that

$$R_{MSto1}(h) = \sum_{k=2}^{k=n} R_{kto1}(h), \qquad (4)$$

The majority of results of this work are shown and discussed below in terms of ratios $R_{MSto1}(h)$ and $R_{2to1}(h)$.

In order to discuss our results in terms used in the literature (see, e.g. Young and Vaughan, 2009; Young et al., 2013; Vaughan et al., 2009; Garnier et al., 2015; and references therein), we employ the following notations and relationships. The

"attenuated backscatter", i.e. lidar signals $S_{MS}(h)$ computed in multiple-scattering conditions (corrected for the offset, instrumental factors, and the two-way ozone transmittance) is expressed as (see e.g. Young et al., 2013)

$$S_{MS}(h) = [\beta_p(h) + \beta_m(h)] \cdot T_m^2(h) \cdot T_{pA}^2(h), \qquad (5)$$

where $T_{pA}^2(h) = \exp[-2\eta(h_b, h)\tau_p(h_b, h)]$ is the apparent particulate two-way transmittance; $\eta(h_b, h)$ is the multiple scattering function (see Appendix in Shcherbakov et al. (2022) for details), i.e. a parameterization describing the effect of MS

on particulate extinction (Young et al., 2013).

$$S_m(h) = [\beta_m(h)] \cdot T_m^2(h) \qquad (6)$$

is a lidar signal computed under the condition of the free molecular atmosphere using available meteorological data (see e.g. Winker et al., 2009).





The scattering ratio $\mathcal{R}(h) = [\beta_p(h) + \beta_m(h)]/\beta_m(h) = [\beta_p(h)/\beta_m(h)] + 1$ is a useful parameter to work with lidar data;

$\mathcal{R}(h) = 1$ for the molecular atmosphere. The attenuated scattering ratio can be defined as (see e.g. Winker et al., 2009)

$$\mathcal{R}'_{MS}(h) = S_{MS}(h)/S_m(h) = \mathcal{R}(h) \cdot T^2_{pA}(h). \tag{7}$$

The definitions of a MS lidar signal $S_{MS}(h)$ with Eq. (5), of a SS lidar signal $S_1(h)$ with Eq. (1), and of the attenuated scattering

ratio with Eq. (7) lead to the following properties for intervals of the cloud-free molecular atmosphere

$$\mathcal{R}'_{MS}(h) = T^2_{pA}, \tag{8}$$

$$S_1(h) = [\beta_m(h)] \cdot T^2_m(h) \cdot T^2_p = S_m(h) \cdot T^2_p, \tag{9}$$

$$S_{MS}(h) = [\beta_m(h)] \cdot T^2_m(h) \cdot T^2_{pA} = S_m(h) \cdot T^2_{pA}, \tag{10}$$

where $T^2_p = T^2_{pA} = 1$ when $h < h_b$; $T^2_p = const$ and $T^2_{pA} = const$ when $h > h_{end}$, where $h_{end}$ is the distance to the cloud far

edge.

It follows from the relationships above that

$$R_{MSto1}(h) = 0, \mathcal{R}'_{MS}(h) = 1, \tag{11}$$

when $h < h_b$, and

$$R_{MSto1}(h) = \left(T^2_{pA}/T^2_p\right) - 1 = const, \mathcal{R}'_{MS}(h) = T^2_{pA} = const. \tag{12}$$

when $h > h_{end}$.

In addition, the following relationship can be useful for interpretation of Monte-Carlo data

$$\mathcal{R}'_{MS}(h) = [R_{MSto1}(h) + 1] \cdot T^2_p = const \text{ when } h > h_{end}. \tag{13}$$

It should be underlined that, if measurement errors are neglected, the functions $R_{MSto1}(h)$ and $\mathcal{R}'_{MS}(h)$ are expected to be

constant when we are dealing within the intervals of the cloud free molecular atmosphere, i.e. either $h < h_b$ or $h > h_{end}$.

Moreover, the apparent optical thickness $\eta \cdot \tau_p(h_b, h_{end})$ of the cloud can be easily computed (see e.g. Garnier et al., 2015)

$$\eta \cdot \tau_p(h_b, h_{end}) = -0.5 \cdot ln[\mathcal{R}'_{MS}(h_2)/\mathcal{R}'_{MS}(h_1)] \tag{14}$$

where $h_1$ and $h_2$ can be any points satisfying the conditions $h_1 < h_b$ and $h_2 > h_{end}$.

**2.2 Simulation software and conditions**

Our tool to perform Mont-Carlo simulations of lidar signals was the McRALI (Monte-Carlo Radar Lidar) software developed

at the Laboratoire de Météorologie Physique (Szczap et al., 2021; Alkasem et al., 2017). The software employs a forward

Monte-Carlo (MC) approach along with the locate estimates method to simulate propagation of radiation (see e.g. Marchuk et

al., 2013). McRALI is based on the 3DMCPOL model (Cornet et al., 2010). The polarization state of the radiation is computed

using Stokes vectors and scattering matrixes of atmospheric compounds. It takes into account molecular scattering. In this

work, the properties of the atmosphere were assigned according to the 1976 standard atmosphere (NOAA, 1976). McRALI is

a fully 3D software, that is, values of the extinction coefficients, the single scattering albedos, and the scattering matrixes are

assigned in 3D-space. Moreover, the mixture of different types of aerosols and/or clouds is allowed. The position of a lidar





can be anywhere within or outside of the atmosphere, that is, space-borne, airborne, and ground-based measurement conditions can be simulated. A user can assign a lidar beam direction, a receiver field-of-view (RFOV), and a Stokes vector and a divergence of the emitted light.

In this work, MC data were computed so that photons were integrated over the range gate 20 m; i.e. they correspond to photon counting mode. Such small value of the range gate was chosen with the aim to study multiple scattering in details regardless

of the fact that it does not correspond to real lidar systems. In other words, the spatial resolution of our MC data is 20 m. The orders of scattering up to 20 were considered in MC simulations to compute the total multiple scattering. (We have verified that the difference between data obtained with 20 and 10 orders of scattering was not statistically significant for the simulations conditions of this work.)

The results of this work are complementary to the data of the work by Shcherbakov et al. (2022) since most of our MC

simulations were performed for the cases of (*i*) the same water cloud and (*ii*) the same jet-stream cirrus cloud. The corresponding normalized phase function $f_W(\theta)$ and $f_{JSC}(\theta)$, where $\theta$ is scattering angle, are shown in Fig. 1 by blue and black curves, respectively; their behaviour at forward and backward angles can be seen in the insets. The scattering matrix of water cloud was computed according to the Mie theory for water spheres having the gamma size distribution with the effective diameter $d_{eff} = 18.0$ μm (the standard deviation of 5.3 μm). The SS characteristics of cirrus cloud were computed using the

Improved Geometric Optics Method (Yang and Liou, 1996); the size distribution of particles was taken to be the gamma distribution with $d_{eff} = 56.8$ μm (the standard deviation 20.1 μm).

To gain a better understanding of the set of parameters that govern MS effect we will use artificial phase functions. We will use the term "chimerical" for them to underline that they are not expected to fit a real phase function, but to reproduce some of its characteristics. A chimerical phase function is a sum of three Gaussian functions $G_i(\theta), i = 1,2,3$: a narrow peak $G_1(\theta)$

for scattering angles close to 0° (forward direction), a somewhat smoothly varying function $G_2(\theta)$, and a peak $G_3(\pi - \theta)$ for angles close to 180°. (We use the notation "$\pi - \theta$" in order to underline that we are dealing with the peak at the backward direction). The first two component are in accord with the work by Weinman (1968). The third component is inspired by the results of the work (Zhou and Yang, 2015) where rigorous numerical simulations based on solving Maxwell's equations showed that a backscattering peak exists even in cases of randomly oriented ice crystals. We use the formula $G_i(\theta) = a_i \cdot$

$exp[-(\theta^2)/(\theta_{s,i}^2)]$ to define the Gaussian components $G_i(\theta)$, where $\theta_{s,i}$ is the $1/e$ angular half-width. The utility of a chimerical phase function consists in the possibility to vary one of parameters whereas other characteristics remain unchanged. The first chimerical phase function $f_{Ch1}(\theta) = \sum_{i=1}^{3} G_i(\theta)$ is shown by the red curve in Fig. 1. It was designed to meet the following properties of the normalized phase function $f_W(\theta)$ of the water cloud. $f_{Ch1}(0) = f_W(0)$, $f_{Ch1}(\pi) = f_W(\pi)$, and the both phase functions have the same value of the lidar ratio. The width of the Gaussian component $G_1(\theta)$ of $f_{Ch1}(\theta)$ is assigned

so that the function $f_{Ch1}(\theta)\sin(\theta)$ has the maximum at the same value $\theta_{max} = 10.82$ mrad as the function $f_W(\theta)\sin(\theta)$ has. The width of $G_3(\pi - \theta)$ was adjusted so that $f_{Ch1}(\theta)$ and $f_W(\theta)$ lead to coincident functions $R_{2to1}(h)$ (see Section 3.1). The Gaussian component $G_2(\theta)$ is large; it assures that sufficient proportion of photons is scattered sideward. The chimerical phase





function $f_{Ch2}(\theta)$ is shown by the green curve in Fig. 1. It has the same components $G_1(\theta)$ and $G_2(\theta)$ as $f_{Ch1}(\theta)$. The only difference is $G_3(\pi - \theta)$, which is larger by a factor of 15. Consequently, the curves of $f_{Ch1}(\theta)$ and $f_{Ch2}(\theta)$ coincide in Fig. 1

till the scattering angle about 170°. The values of parameters $a_i$ and $\theta_{s,i}$ of the chimerical phase functions are given in Table 1.

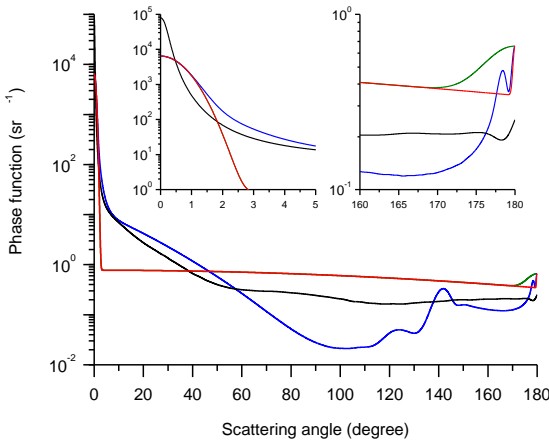

**Figure 1.** Normalized phase functions: water cloud – blue lines, jet-stream cirrus – black lines, chimerical phase function $\boldsymbol{f_{Ch1}(\theta)}$ – red lines, chimerical phase function $\boldsymbol{f_{Ch2}(\theta)}$ – green lines.

**Table 1.** Values of the input parameters of the chimerical phase functions.

| | $a_1$ | $a_2$ | $a_3$ | $\theta_{s,1}$ (degree) | $\theta_{s,2}$ (degree) | $\theta_{s,3}$ (degree) |
|---|---|---|---|---|---|---|
| $f_{Ch1}(\theta)$ | 2.2690 | 0.062 | 4.086E-05 | 0.88 | 200.0 | 0.327 |
| $f_{Ch2}(\theta)$ | 2.2712 | 0.062 | 6.26045E-04 | 0.88 | 200.0 | 5.0 |

Along with MC data we provide the results, which we obtained using our codes based on the Eloranta model (EM) (Eloranta, 1998) (see details in Appendix). In what follows the data of our simulations using the EM are referred as "multiple scattering" or are labelled as "MS" when up to 5 orders of scattering were considered. The contribution of the fifth order is sufficiently

small in all cases considered below. The addition of the sixth order increases unreasonably the time of computing with the EM.

In this work, each cloud layer is plane-parallel (unless otherwise stated) and homogeneous; all scattering matrixes were assigned with an angular resolution of 0.01° (about 0.175 mrad); the emitter wavelength $\lambda$ is 0.532 μm. Almost all MC simulations were performed for cloud particles having the extinction coefficient $\varepsilon_p(h) = 1.0$ km$^{-1}$ for the following reasons.

On the one hand, technical capacities of contemporary lidars provide possibility to record signals from the cloud free





atmosphere beyond the far edge of a cloud having the optical thickness $\tau_p = 3.0$. On the other hand, MS effect cannot be neglected and is clearly seen in a number of cases (Shcherbakov et al., 2022).

## 3 Ground-based lidar

The results of this section were obtained for a ground-based lidar, which is at the altitude $H = 0$ km, and the distance to the
clouds base is 8 km. The full receiver field-of-view (RFOV) is 1.0 mrad (except Figs. 3c and 3d); the full emitter field-of-view (EFOV) is 0.14 mrad. The emitter wavelength $\lambda$ is 0.532 μm. (We used the characteristics of the lidar system that is in operation at Clermont-Ferrand (Freville et al., 2015).) In order to assure good statistical quality of our Monte-Carlo modelling, each signal was simulated with $2 \cdot 10^{11}$ photons emitted by the lidar (with $4 \cdot 10^{11}$ photons for the cirrus clouds). Simulations of signals were performed for the orders of scattering $n = 1$ (single scattering), $n = 2$ (double scattering), and multiple
scattering with $n$ equal 20.

### 3.1 Single-layer cloud

The single-layer cloud is within the altitude range $H \in[8., 11.]$ km, that is, has the optical thickness $\tau_p = 3.0$ (the extinction coefficient $\varepsilon_p = 1.0$ km⁻¹). Black and red points in Fig. 2 show the results of our MC simulations reported in terms of the ratios $R_{MSto1}(d)$ and $R_{2to1}(d)$, respectively. The parameter $d$ is the distance measured from the cloud base, i.e. the cloud is within
the range $d \in[0., 3.]$ km. The interval $d \in]3., 6.]$ km is the cloud-free molecular atmosphere, i.e. $H \in]11., 14.]$ km. Figures 2a and 2b correspond to the water and cirrus cloud, respectively. The relative contributions $R_{MSto1}(d)$ and $R_{2to1}(d)$ computed using the EM are shown in Fig. 2 by the green and blue lines, respectively. The MC in-cloud data, i.e. $d \in[0., 3.]$ km, were reported and discussed in the work (Shcherbakov et al., 2022). The focus of interest for this work is the cloud-free molecular atmosphere, that is, the interval $d \in]3., 6.]$ km.

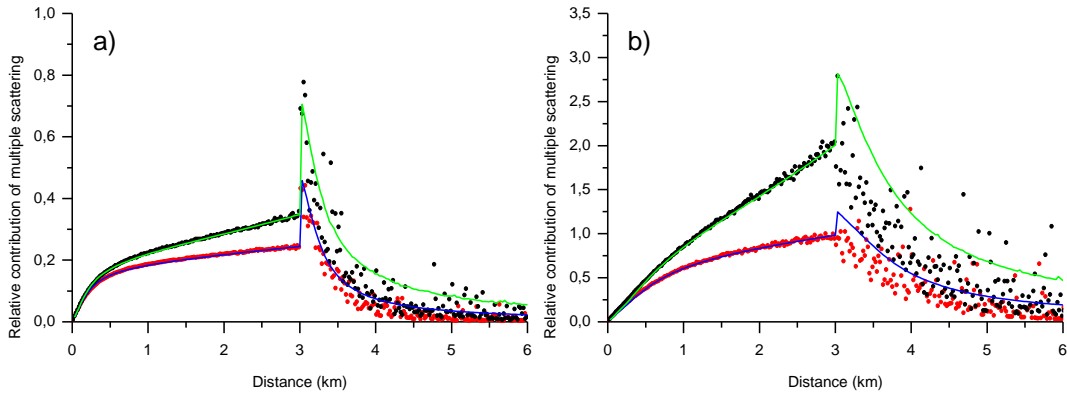






**Figure 2.** Monte Carlo simulations of multiple scattering $R_{MSto1}$ (black points) and double scattering $R_{2to1}$ (red points) relative contributions to lidar signals; (a) water cloud, (b) jet-stream cirrus. Eloranta-model simulations are shown by the green (MS) and the blue (double scattering) curves.

Our MC simulations reveal the following features. The function $R_{MSto1}(d)$ and $R_{2to1}(d)$ have undergone a stepwise jump immediately beyond the cloud far edge. It should be kept in mind that lidar signals $S_1(h)$, $S_2(h)$, and $S_{MS}(h)$ sharply decrease immediately beyond the cloud far edge (see e.g. Fig. 7 below). The decreasing of $S_1(h)$ is faster, therefore the ratios $R_{MSto1}(d)$ and $R_{2to1}(d)$ have undergone a stepwise jump. After the jump, the ratios decrease and tend to zero. It should be noted that the stepwise jump with subsequent decreasing were already reported in the work (Flesia and Starkov, 1996), where multiple

scattering to a space-borne lidar return from clear molecular atmosphere obscured by transparent upper-level crystal clouds was assessed by the use of the Monte Carlo technique.

It is seen in Figs. 2a and b that the EM is able to well simulate $R_{MSto1}(d)$ and $R_{2to1}(d)$ within the cloud layer $d \in [0., 3.]$ km. The simulation results can be considered as acceptable in the range $d \in ]3., 6.]$ km. That is, our EM data show the stepwise jump of the same amplitude as the MC data immediately beyond the cloud far edge. In contrast, the decreasing rate of

$R_{MSto1}(d)$ and $R_{2to1}(d)$ is lower. Therefore, the EM slightly overestimates the multiple scattering contribution in the cloud-free molecular atmosphere. We achieved the good agreement with the MC data by adjusting the EM parameters only for the interval $d \in [0., 3.]$ km (see details and the definition of parameters in Appendix). We have not succeeded to improve the fitting quality for the range $d \in ]3., 6.]$ km by varying the fraction $\gamma(h)$ of the energy in the forward peak of the phase function. Thus, all EM data of this work were obtained with $\gamma(h) = 1/2$. The key fact is that the fittings of the water cloud case were obtained

using values of the ratios $\mathcal{P}_n(\pi, h)/\mathcal{P}_1(\pi, h)$, $n = 2, \ldots, 5$ about 0.5 or lower (see discussion in Appendix). As for the cloud-free molecular atmosphere, the ratios $\mathcal{P}_n(\pi, h)/\mathcal{P}_1(\pi, h)$ are "equal to 1.0 due to the broad nature of the molecular phase function near the backscatter direction" (Eloranta, 1998; Whiteman et al., 2001).

### 3.1.1 Stepwise jump and escape effect

The properties of $R_{2to1}(d)$, i.e. of double scattering, within the interval $d \in ]3., 6.]$ km cannot be due to the stretching of the

pulse length. The definition and a good explanation of the pulse stretching can be found in (Miller and Stephens, 1999). The simplified explanation could be as follows. In the case of double scattering, a photon goes through two scattering within the cloud, returns to the receiver and has the round-trip distance equal to the case of the single scattering from the range of the free atmosphere beyond the cloud far edge. The maximal round-trip distance depends on the configuration geometry, i.e. on the distance from the lidar to the cloud, the cloud depth, the EFOV, and the RFOV. In the case of Fig. 2, the EFOV and the RFOV

are narrow whereas the distance from the lidar to particles of the cloud layer is quite low. The round-trip distance of a double-scattered photon can gain a few meters in such conditions. It means that only the range $d \in ]3., 3.02]$ km of $R_{2to1}(d)$ can be somewhat affected by the pulse-length stretching. Thus, it is safe to assume that the stepwise jump of $R_{MSto1}(d)$ and $R_{2to1}(d)$





is due to the stepwise jump in phase-function properties at angles close to 180° (the phase function of particles within the cloud

and the Rayleigh scattering within the free atmosphere).

That assumption is confirmed by the plots in Figs. 3a and b, where the results of MC simulations are shown as the ratios

$R_{MSto1}(d)$ and $R_{2to1}(d)$ for the cases of the chimerical phase functions $f_{Ch1}(\theta)$ and $f_{Ch2}(\theta)$, respectively. We adjusted the

width of the component $G_3(\pi - \theta)$ of $f_{Ch1}(\theta)$ so that MC simulations with $f_{Ch1}(\theta)$ give the same $R_{2to1}(d)$ within the range

$d \in[0., 3.]$ km as $R_{2to1}(d)$ of the water cloud in Fig. 2a. It turns out that the peak at the backward direction of the water cloud

is a little bit larger than $G_3(\pi - \theta)$ of $f_{Ch1}(\theta)$ (see Fig. 1). Nevertheless, the MC data in Fig. 3a are coincident with the MC

data in Fig. 2 for the ratios $R_{MSto1}(d)$ and $R_{2to1}(d)$ within the full range $d \in[0., 6.]$ km. The good agreement is due to the fact

that one of the key parameters of multiple scattering is a weighted average of a phase function near the backscatter direction

(see Eq. (10) of Eloranta, 1998). That average depends on properties of the forward and backward peaks of the phase function

as well as the receiver field of view. It means that we fitted the value of the weighted average when we adjusted the width of

the Gaussian function $G_3(\pi - \theta)$.

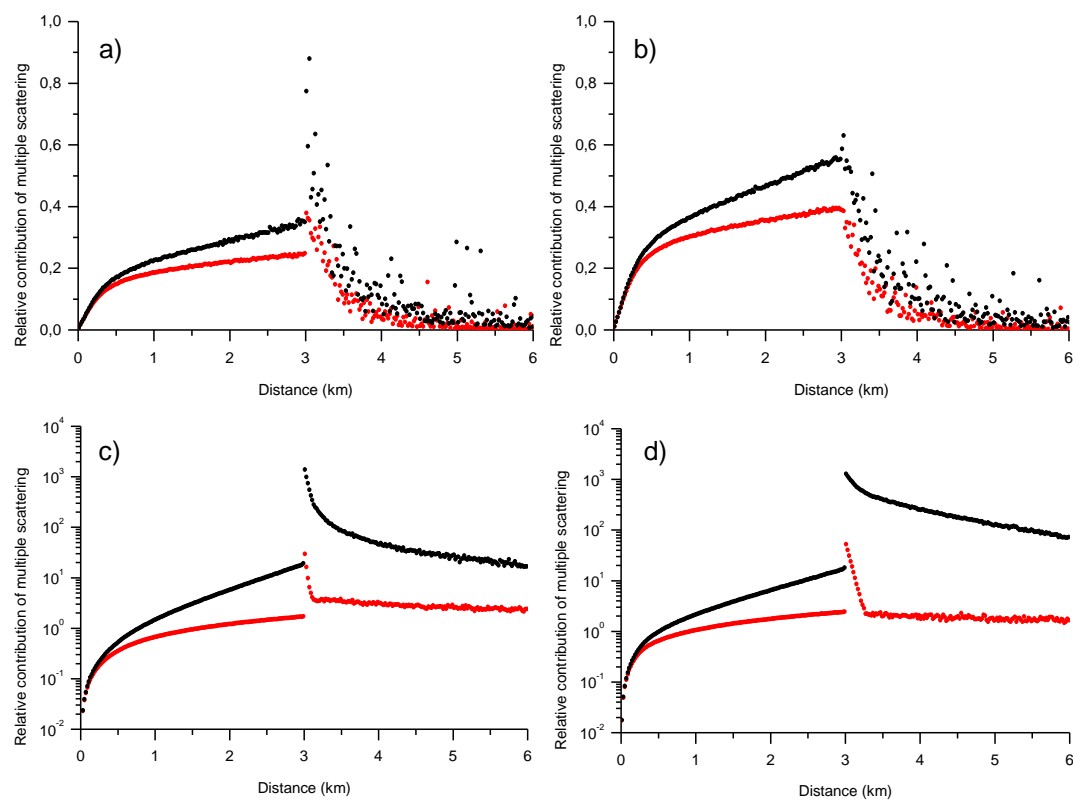




**Figure 3.** Monte Carlo simulations of multiple scattering $R_{MSto1}$ (black points) and double scattering $R_{2to1}$ (red points) relative contributions to lidar signals; (a) and (b) RFOV=1 mrad; (c) and (d) RFOV=110 mrad; (a) chimerical phase function $f_{Ch1}(\theta)$; (b) and (d) chimerical phase function $f_{Ch2}(\theta)$; (c) water cloud.

The chimerical phase functions $f_{Ch1}(\theta)$ and $f_{Ch2}(\theta)$ have the same values of the parameters $\theta_{s,1}$ and $\theta_{s,2}$, but the width $\theta_{s,3}$ is 15 times higher for $f_{Ch2}(\theta)$. That leads to marked distinctions in the ratios $R_{MSto1}(d)$ and $R_{2to1}(d)$ (see Fig. 3b). There is no a stepwise jump immediately beyond the cloud far edge. It means that the component $G_3(\pi - \theta)$ of $f_{Ch2}(\theta)$ is large enough to have the weighted average equal to $f_{Ch2}(\pi)$ as in the case of the Rayleigh phase function. To put it differently, a higher proportion of photons is scattered in the backward direction within the RFOV and contribute to lidar signals. As a consequence,

the ratios $R_{MSto1}(d)$ and $R_{2to1}(d)$ of Fig. 3b are much higher than in Figs. 2a and 3a for the in-cloud range $d \in [0., 3.]$ km. The MC data of Figs. 2a, 3a, and 3b are coincident within the interval of the cloud-free molecular atmosphere $d \in ]3., 6.]$ km if they are superimposed in a same figure. Thus, it is safe to assume that two following properties are defined by the forward-peak width of the phase function and the RFOV: (i) the proportion of scattered photons emerging from a cloud and subsequently scattered in the backward direction to the receiver, and (ii) the decreasing rate of the relative contribution of

multiple scattering outside the cloud. (We recall that we are dealing with a ground-based lidar, the cloud optical-thickness is not high, and the RFOV and the EFOV are narrow.) That assumption explains the difference in the decreasing rates within the range $d \in [3., 6.]$ km between Figs. 2a and 2b. The forward peak of the scattering phase function of the cirrus cloud is much stronger compared to that one of the water cloud. That leads to the slower decreasing rate.

        In our opinion, those properties are a direct consequence of the fact that the photons scattered within the forward peak escape

the sampling volume. (For simplicity sake, we will use the term "escape effect" to refer to that phenomenon.) The following arguments justify that statement. Let us consider the case of the forward scattering. If we refer to Fraunhofer diffraction by large spheres (see e.g. Ch. 8.3 of Van de Hulst, 1981), the first dark ring, i.e. the Airy disk, is at the angle $\theta_{Airy} \approx 36$ mrad (about 2 degrees) when a particle has the diameter 18 μm and the wavelength $\lambda$ is 0.532 μm. $\theta_{Airy} \gg$ RFOV/2 = 0.5 mrad. If we refer to the phase function of the water cloud (see Fig. 1), the intervals of angles [0., 0.5] mrad and [0., 36.] mrad account

for 0.059% and 41.3% of the total scattered light, respectively. In such conditions, it can be hardly expected that a large proportion of small angle forward-scattered photons always remain within the field of view of the detector; more likely photons escape the sampling volume.

        With the aim of confirming our arguments, we performed MC simulations for the cases of larger RFOVs. All but the RFOV parameters of lidar configuration and the water cloud properties are the same as in the case of Fig. 2a. The full RFOV is 110

mrad. Such large value assures that the narrow peak $G_1(\theta)$ (forward direction) of the chimerical phase functions is totally within the angle RFOV/2 = 55 mrad (about 3 degrees). Black and red points in Fig. 3c and d show the results of our MC simulations reported in terms of the ratios $R_{MSto1}(d)$ and $R_{2to1}(d)$, respectively. Figures 3c and 3d show the cases of the water cloud and the chimerical phase function $f_{Ch2}(\theta)$, respectively. (Note the log scale of the y-axis in Figs. 3c and d.)



It is seen that the ratios $R_{2to1}(d)$ have undergone a very strong stepwise-jump immediately beyond the cloud far edge. If we

consider the configuration geometry, i.e. the angle RFOV/2 = 55 mrad and the distance from the lidar to the cloud far edge

of 11 km, the pulse stretching is the cause of the jump. The pulse stretching strongly affects $R_{2to1}(d)$ in Fig. 3d within the

range $d \in ]3., 3.25]$ km. The ratio $R_{2to1}(d)$ is almost constant within the interval $d \in [3.25, 6.]$ km. (In our opinion, the slight

decreasing is due to the pulse-stretching effect.) The properties of the ratio $R_{2to1}(d)$ in Fig. 3c are the same but less

pronounced. We can hypothesize that photons that were forward scattered within angles > 3 degrees play noticeable role in

the case of the water cloud. The results shown in Figs. 3c and 3d mean that only when the RFOV is sufficiently large, small

angle forward-scattered photons remain within the field of view of the detector and the escape effect is eliminated. As for the

cases of multiple scattering, the ratios $R_{MSto1}(d)$ in Figs. 3c and 3d are very high due to the large RFOV and the escape effect

is evident.

It is seen that the decreasing due to the escape effect is an inherent part of $R_{MSto1}(d)$ properties within the free atmosphere

beyond the cloud far edge. That property is in direct contradiction with Eqs. (10 -14) above, which are the consequences of

Eq. (5). Therefore, we can conclude that Eq. (5) is an approximate model of lidar signals under MS conditions because it does

not take into account the escape effect.

### 3.1.2 Practical aspects

Knowing that the value of the extinction coefficient $\varepsilon_p = 1.0$ km$^{-1}$ is not typical of cirrus clouds, we provide practitioners with

estimations of MS effects for lower values of $\varepsilon_p$ of the jet-stream cirrus. Black points in Fig. 4 show the results of our MC

simulations reported in terms of the ratio $R_{MSto1}(d)$. The parameter $d$ is the distance measured from the cloud base, i.e. the

cloud is within the range $d \in [0., 3.]$ km. The interval $d \in ]3., 6.]$ km is the cloud-free molecular atmosphere, i.e. the altitude

$H \in ]11., 14.]$ km. The MC simulations were performed for the same geometry of plane-parallel homogeneous cloud as in the

case of Fig. 2b. The only difference consists in the values of the extinction coefficient, which were $\varepsilon_p = 0.06$ and $\varepsilon_p = 0.2$

km$^{-1}$ in Figs. 4a and 4b, respectively. It is seen that the lidar data are affected by MS within the cloud ($d \in [0., 3.]$ km) and

above the cloud far edge ($d \in ]3., 6.]$ km). The cases of Fig. 4 are characterized by the quite low values of the extinction

coefficient, i.e. the low probability of the interaction of a photon with cloud particles. Moreover, the RFOV, i.e. the sampling

volume, and the forward peak of the scattering phase function are narrow. All that led to the rather high dispersion of the MC

data in Fig. 4 despite the very high number $4 \cdot 10^{11}$ of sampled photons. Nevertheless, it is safe to conclude that the MS

contribution is decreasing with the distance from the cloud for edge due to the escape effect. The MS contribution is lower

than 5% within the regions of the cloud-free molecular atmosphere having the distance from the cloud far edge about 1 km or

higher when and $\varepsilon_p \leq 0.2$ km$^{-1}$.

Table 2 shows the values of the apparent optical thickness $\eta \cdot \tau_p(h_b, h_{end})$ of the cloud computed using Eq. (14) with two

values of $h_2$ assigned within the interval $d \in ]3., 6.]$ km (the altitude $H \in ]11., 14.]$ km). The values 0.2 and 3.0 km mean the

distance from the cloud far edge within the cloud-free molecular atmosphere, they correspond to $d = 3.2$ ($H = 11.2$) and $d =$




6.0 ($H = 14.0$) km, respectively. It is seen that the computed value of the apparent optical thickness depends on the chosen value of $h_2$. It is close to the value of the real optical thickness when $h_2 = 14.0$ km and can be about 15% lower when $h_2 = 11.2$ km in the cases of the jet-stream cirrus.

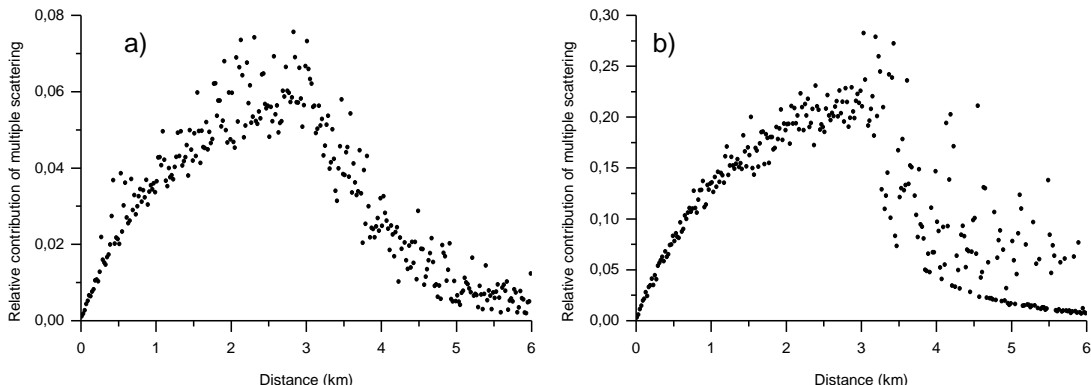

**Figure 4.** Monte Carlo simulations of multiple scattering relative contributions $R_{MSto1}$ to lidar signals from the jet-stream cirrus; the extinction coefficient is (a) 0.06 km$^{-1}$ and (b) 0.2 km$^{-1}$.

**Table 2.** Estimated values of the apparent optical thickness $\eta \cdot \tau_p(h_b, h_{end})$.

| | | Apparent optical thickness | | Real optical thickness |
|---|---|---|---|---|
| Distance from cloud (km)<br>MC data case (figure) | | 0.2 | 3.0 | |
| Water cloud | (Fig. 2a) | 2.83 | 2.99 | 3.0 |
| Jet-stream cirrus | (Fig. 2b) | 2.58 | 2.94 | 3.0 |
| Jet-stream cirrus | (Fig. 4a) | 0.15 | 0.18 | 0.18 |
| Jet-stream cirrus | (Fig. 4b) | 0.51 | 0.60 | 0.60 |

## 3.2 Two-layered cloud

The two-layered cloud consists of homogeneous layers situated at the altitudes from 8 to 9 km and from 10 to 11 km. Every layer has the optical thickness $\tau_p = 1.0$; the total optical thickness is $\tau_p = 2.0$. The black and red points in Fig. 5 show the results of our MC simulations reported in terms of the ratios $R_{MSto1}(d)$ and $R_{2to1}(d)$, respectively. The parameter $d$ is the distance measured from the cloud base, i.e. the layers are within the ranges $d \in [0., 1.]$ and $d \in [2., 3.]$ km. The intervals $d \in ]1.,$



2.[ and $d \in$]3., 6.] km are the cloud-free molecular atmosphere. Figures 5a and 5b correspond to the water and cirrus cloud, respectively. The relative contributions $R_{MSto1}(d)$ and $R_{2to1}(d)$ computed using the EM are shown in Fig. 5 by green and blue lines, respectively.

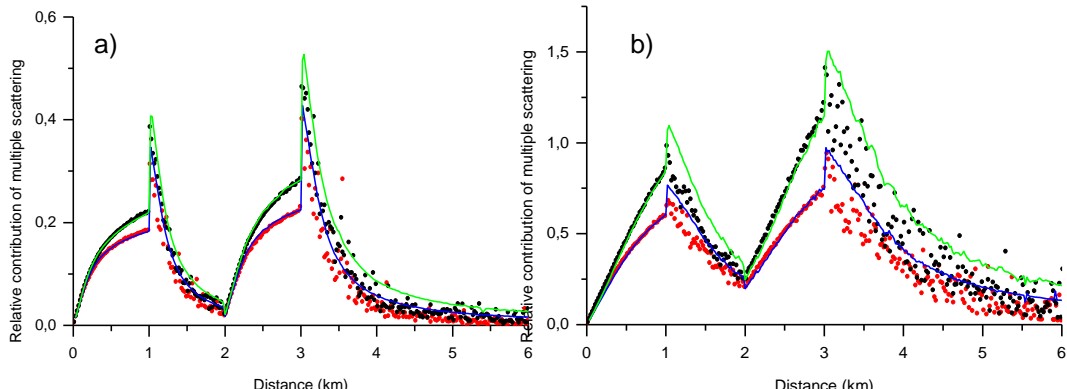

**Figure 5.** Monte Carlo simulations of multiple scattering $R_{MSto1}$ (black points) and double scattering $R_{2to1}$ (red points)
relative contributions to lidar signals; (a) water cloud, (b) jet-stream cirrus. Eloranta-model simulations are green (MS) and blue (double scattering) curves.

The features of the MC data within the intervals of the cloud-free molecular atmosphere are closely similar to those addressed in the previous section. In addition, the following property is noteworthy. The decreasing rate of the MS relative contributions
in the case of the water cloud is quite fast and therefore scattered photons emerging from the first layer almost do not affect the lidar signal from the second layer. (The values of the ratios $R_{MSto1}(d = 2.0)$ and $R_{2to1}(d = 2.0)$ are close to zero.) As it was discussed above, the decreasing rate of the MS relative contributions in the case of the cirrus clouds is much slower compared to the water clouds. Therefore, scattered photons emerging from the first layer affect the lidar signal from the second layer so that the ratios $R_{MSto1}(d = 2.0)$ and $R_{2to1}(d = 2.0)$ are about 0.25. In other words, the lidar signal from the near edge
of the second layer is about 25 % higher compared to the SS approximation. Obviously, the distance between cloud layers is a key parameter. The values of the ratios $R_{MSto1}(d)$ and $R_{2to1}(d)$ at the near edge of the second layer vary if that distance changes.

As for our EM simulations, we used the same values of the ratios $\mathcal{P}_n(\pi, h)/\mathcal{P}_1(\pi, h)$, $n = 2, ..., 5$ as in the previous section for each cloud layer. We can conclude another time that the EM is able to well simulate MS contributions within the cloud
layers. As for the intervals of the cloud-free molecular atmosphere, the behavior of $R_{MSto1}(d)$ and $R_{2to1}(d)$ is correct whereas the MS contribution is slightly overestimated.



## 4 Space-borne lidar

The results of this section were obtained for a space-borne lidar, which is at the altitude $H = 705$ km. The full receiver field-of-view (RFOV) is 0.13 mrad; the full emitter field-of-view (EFOV) is 0.1 mrad. The emitter wavelength $\lambda$ is 0.532 µm. Those

values correspond to the technical characteristics of the Cloud-Aerosol Lidar with Orthogonal Polarization (CALIOP) (Young and Vaughan, 2009). In order to assure good statistical quality of our Monte-Carlo modelling, each signal was simulated with $2 \cdot 10^{11}$ photons emitted by the lidar (with $4 \cdot 10^{11}$ photons for the cirrus clouds). Simulations of signals were performed for the orders of scattering $n = 1$ (single scattering), $n = 2$ (double scattering), and multiple scattering with $n$ equal 20.

### 4.1 Single-layer cloud

#### 4.1.1 Plane-parallel homogeneous cloud

The single-layer cloud is within the altitude range $H \in [8., 11.]$ km and has the optical thickness $\tau_p = 3.0$ (the extinction coefficient $\varepsilon_p(h) = 1.0$ km$^{-1}$). Black and red points in Fig. 6 show the results of our MC simulations reported in terms of the ratios $R_{MSto1}(d)$ and $R_{2to1}(d)$, respectively. The parameter $d$ is the distance measured from the cloud near edge from the lidar, i.e. the cloud is within the range $d \in [0., 3.]$ km. ($d = 0.$ km corresponds to the altitude $H = 11.$ km.) The interval $d \in ]3.,$

11.] km (the altitude $H$ from 8. to 0. km) is the cloud-free molecular atmosphere below the cloud. Figures 6a and 6b correspond to the water and cirrus cloud, respectively. The relative contributions $R_{MSto1}(d)$ and $R_{2to1}(d)$ computed using the EM are shown in Fig. 6 by the green and blue lines, respectively. The MC in-cloud data, i.e. $d \in [0., 3.]$ km, were reported and discussed in the work (Shcherbakov et al., 2022). The focus of interest for this work is the cloud-free molecular atmosphere, that is, the interval $d \in ]3., 11.]$ km.

The features of the MC data within the range of the cloud-free molecular atmosphere have much in common with those addressed in Section 3.1. In addition, the following properties are noteworthy. In the case of the space-borne lidar, the difference between values of $R_{MSto1}(d)$ and $R_{2to1}(d)$ is much larger compared to Figs. 2a and b, that is, the third and higher orders of scattering dominate. The escape effect does affect the lidar signals. At the same time, the decreasing rate of $R_{MSto1}(d)$ and $R_{2to1}(d)$ is much slower. The distance 8 km from the cloud is not sufficient to reach the conditions of the SS

approximation. That result is of importance for practitioners, who work with data of space-borne lidars, because the estimated value of the apparent optical thickness depends on the distance from the cloud far edge chosen as the reference point $h_2$ (see details in Section 4.1.3).

As in the cases of the ground-based lidar, the EM is able to well simulate MS contributions within the cloud layers. We employed the same approach to obtain values of the ratios $\mathcal{P}_n(\pi, h)/\mathcal{P}_1(\pi, h)$, $n = 2, ..., 5$ as in Section 3.1 (see Appendix for

details). As in the cases of the ground-based lidar, the fittings of the water cloud case were obtained using values of the ratios $\mathcal{P}_n(\pi, h)/\mathcal{P}_1(\pi, h)$, $n = 2, ..., 5$ about 0.5 or lower. The behavior of $R_{MSto1}(d)$ and $R_{2to1}(d)$ is correct within the range of the cloud-free molecular atmosphere, whereas the MS contribution is slightly overestimated. It is seen that the EM is not able to





reproduce the amplitude of the stepwise jump of $R_{MSto1}(d)$ immediately beyond the cloud far edge. Thus, the stepwise jump in those cases is not only due to the stepwise jump in phase-function properties for angles close to $180°$. We can suggest that

the range $d \in ]3., 3.1]$ km is somewhat affected by the pulse stretching.

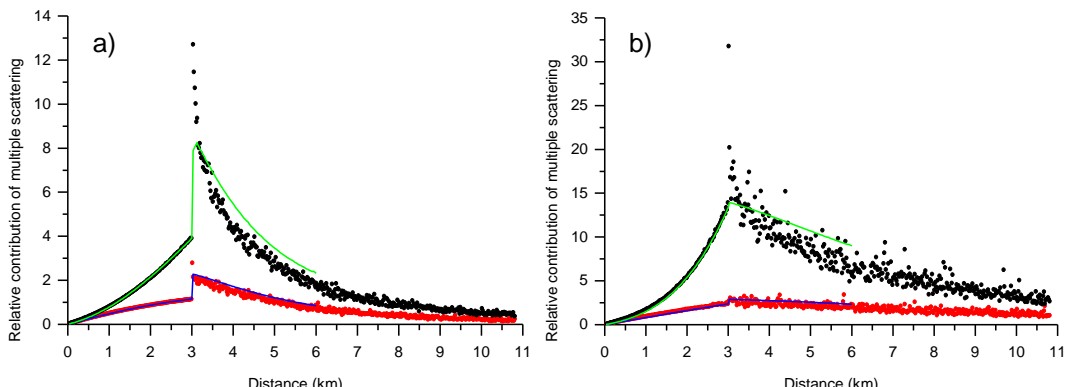

**Figure 6.** Monte Carlo simulations of multiple scattering $R_{MSto1}$ (black points) and double scattering $R_{2to1}$ (red points) relative contributions to lidar signals; (a) water cloud, (b) jet-stream cirrus. Eloranta-model simulations are green (MS) and blue (double scattering) curves.

**4.1.2 Non-uniform beam filling**

Another problem associated with a reference range taken below the cloud base consists in the following. Space-borne lidars such as CALIOP have a quite large laser footprint, so that a cloud field can be horizontally heterogeneous within the footprint. That raises the well-known problem of non-uniform beam filling (NUBF). Some effects of the NUBF on lidar data within cloud fields were addressed in works (Szczap et al., 2021; Alkasem et al., 2017). The lidar signals from the cloud-free

atmosphere below cloud base are affected as well.

In order to understand some basic properties under 3D multiple-scattering conditions, we performed simulations for the case of a very simple 3D field. (All data of this subsection were obtained using the McRALI software, i.e. the Monte Carlo method.) A homogeneous cloud covers half of the field as viewed from the top. The centre of CALIOP laser footprint is exactly on the cloud border. In other words, one half of the laser beam passes through the cloud, and the other half goes through the molecular-

atmosphere. The thickness of the cloud is 3 km and the cloud top altitude is 11 km. The extinction coefficient of particles of the cloud is 2 km$^{-1}$. To put it differently, we have chosen the value of the extinction coefficient so that the amount of particles within the volume bounded by the EFOV in the 3D case is exactly the same as in the case of the plane-parallel homogeneous cloud. The same is true if we consider the sampling volume. Therefore, the optical thickness of the 3D cloud, when averaged over the EFOV, is $3. = (0. + 6.)/2$.





The NUBF effect is so high in such conditions that it has to be shown in terms of lidar signals. Figure 7 is complementary to Fig. 6, i.e. it shows the results based on the same MC simulations but as lidar signals. Figures 7a and 7b correspond to the water and cirrus cloud, respectively. The black and red lines are lidar signals (corrected for the offset and instrumental factors) obtained with the Monte Carlo method in MS conditions and the SS approximation, respectively. We recall that the black and red lines were obtained for the case of the homogeneous plane-parallel cloud having the optical thickness $\tau_p = 3.0$. It is

noteworthy that general features of MS and SS data of Fig. 7 have much in common with results published in the literature (see Fig. 1 in (Donovan, 2016) and Fig. B1 in (Reverdy et al., 2015)). Specifically, (*i*) signals of space-based lidars are affected by MS within the range of the free atmosphere below the cloud base; (*ii*) the MS signals are approaching the SS signals with the increasing distance from the cloud base, i.e. the escape effect is clearly seen.

The green lines in Fig. 7 are the lidar signals computed in the case of the 3D cloud field in MS conditions. Within the in cloud

altitudes $H \in [8., 11.]$ km, they are lower than the MS signals from the corresponding homogeneous cloud (the black lines), which is in total agreement with the theory (see Ch. 3.1 in (Alkasem et al., 2017)). As for the range below the cloud base, it is seen that the NUBF makes the situation much more aggravated. The lidar signals (the green curves) are around 200 times higher compared to the SS approximation for the homogeneous plane-parallel cloud (the red curves). That is the direct consequence of the fact that in the 3D case one half of the laser beam passes through the molecular atmosphere whereas another

half passes through the cloud having $2\tau_p = 6$.

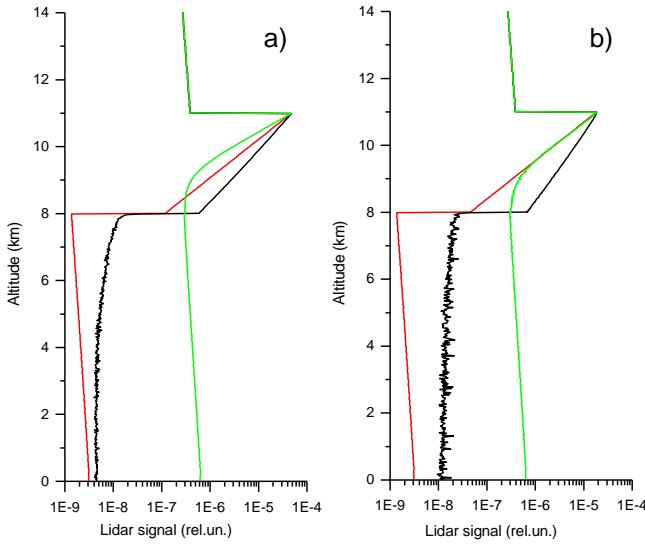

**Figure 7.** Monte Carlo simulations of lidar signals. Homogeneous plane-parallel cloud: (*i*) black curves correspond to multiple scattering and (*ii*) red curves correspond to single scattering conditions; green curves correspond to multiple scattering for 3D cloud field; (a) water cloud, (b) jet-stream cirrus.





### 4.1.3 Practical aspects

Knowing that the value of the extinction coefficient $\varepsilon_p = 1.0$ km$^{-1}$ is not typical of cirrus clouds, we provide practitioners with estimations of the escape and NUBF effects for lower values of $\varepsilon_p$ of the jet-stream cirrus. The black points in Fig. 8 show the values of the MS relative contributions $R_{MSto1}(d)$ (see Eq. 3) in the case of the homogeneous plane-parallel cloud. The MC simulations were performed for the same geometry of plane-parallel homogeneous cloud as in the case of Fig. 6b. The only difference consists in the values of the extinction coefficient, which were $\varepsilon_p = 0.06$ and $\varepsilon_p = 0.2$ km$^{-1}$ in Figs. 8a and 8b, respectively. It is seen that the lidar data are affected by MS within the cloud ($d \in [0., 3.]$ km) and below cloud base ($d \in ]3., 11.]$ km).

NUBF effects are shown by the red points in Fig. 8. As previously, (*i*) a homogeneous cloud covers half of the field as viewed from the top; (*ii*) the centre of CALIOP laser footprint is exactly on the cloud border; (*iii*) the value of $\varepsilon_p$ is by a factor of 2 higher than the extinction coefficient of the corresponding plane-parallel cloud. The values of the extinction coefficient of the 3D cloud, when averaged over the EFOV, are $0.06 = (0. + 0.12)/2$ and $0.2 = (0. + 0.4)/2$ in the cases of Figs. 8a and 8b, respectively. The relative difference

$$R_{NUBFto1}(h) = [S_{NUBF}(h) - S_1(h)]/S_1(h). \tag{15}$$

is taken to be a measure of the NUBF effects, where $S_{NUBF}(h)$ is the MS lidar signal simulated using the Monte Carlo method in the case of the 3D cloud field; $S_1(h)$ is the lidar signal computed in the case of the corresponding plane-parallel cloud under the SS approximation. In other words, the parameter $R_{NUBFto1}$ is devoted to show the relative contribution of the MS and the NUBF taken together.

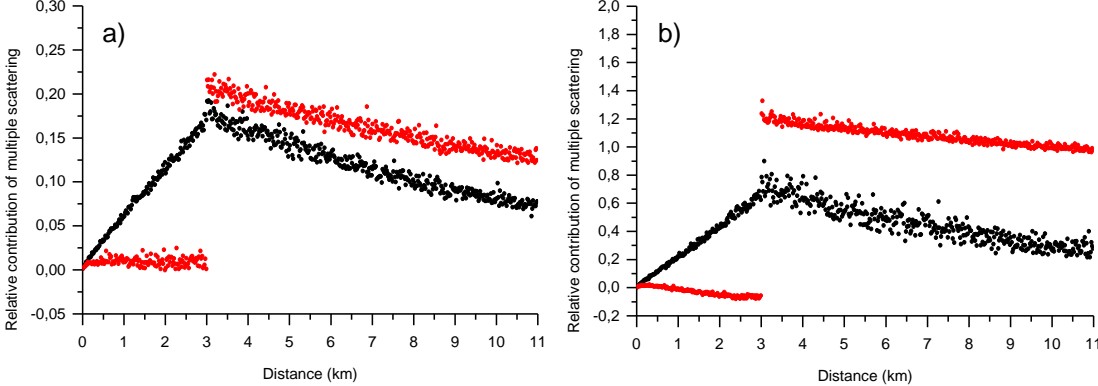

**Figure 8.** Monte Carlo simulations of multiple scattering effects (jet-stream cirrus). Black points are the relative contributions $R_{MSto1}$ to lidar signals from a homogeneous plane-parallel cloud; red points are the relative contributions $R_{NUBFto1}$ of the MS and the NUBF to lidar signals from a 3D cloud; the extinction coefficient is (a) 0.06 km$^{-1}$ and (b) 0.2 km$^{-1}$.





Another time it is seen that the MS effect is somewhat weakened by the NUBF within the cloud range ($d \in$[0., 3.] km). When the optical thickness within the cirrus cloud is quite low, MS lidar signals from a 3D cloud are fairly close to the SS lidar

signals from the corresponding plane-parallel homogeneous cloud (see Figs. 8 and 7b). That property is not valid in the case of the water cloud (see Fig. 7a). It can be hypothesized that more photons escape through the cloud side when the forward-scattering peak of particles is larger.

As for lidar signals within the range of the free atmosphere below cloud base ($d \in$]3., 11.] km), they are always affected either by the multiple scattering or by the non-uniform beam filling. There is no interval where the MS effect is lower than 5%. And,

the NUBF effect on lidar signals is always higher than the MS effect in the case of plane-parallel cloud. We recall that our comparison is done under the condition that the amount of particles within the volume bounded by the EFOV in the 3D case is exactly the same as in the case of the plane-parallel homogeneous cloud.

**Table 3.** Estimated values of the apparent optical thickness $\eta \cdot \tau_p(h_b, h_{end})$. The real optical thickness of 3D clouds is the

value averaged over the EFOV.

| | | Apparent optical thickness | | Real optical thickness |
|---|---|---|---|---|
| Distance from cloud (km) <br> MC data case (figure) | | 0.2 | 7.98 | |
| Water cloud | (Fig. 6a) | 1.91 | 2.81 | 3.0 |
| Jet-stream cirrus | (Fig. 6b) | 1.60 | 2.36 | 3.0 |
| Water cloud (3D) | (Fig. 7a) | 0.32 | 0.35 | 3.0 |
| Jet-stream cirrus (3D) | (Fig. 7b) | 0.31 | 0.34 | 3.0 |
| Jet-stream cirrus | (Fig. 8a) | 0.10 | 0.14 | 0.18 |
| Jet-stream cirrus (3D) | (Fig. 8b) | 0.09 | 0.12 | 0.18 |
| Jet-stream cirrus | (Fig. 8b) | 0.34 | 0.48 | 0.60 |
| Jet-stream cirrus (3D) | (Fig. 8b) | 0.21 | 0.26 | 0.60 |

The escape effect is noteworthy as well. It is seen that $R_{MSto1}(d)$ and $R_{NUBFto1}(d)$ are decreasing with the increasing distance from the cloud base in all four cases of Fig. 8. Examples of its consequences in terms of the apparent optical thickness of the cloud $\eta \cdot \tau_p(h_b, h_{end})$ are given in Table 3. The data were computed using Eq. (14) with two values of $h_2$ assigned within the

interval $d \in$]3., 11.] km (the altitude $H \in$]8., 0.] km). The values 0.2 and 7.98 km mean the distance from the cloud far edge within the cloud-free molecular atmosphere, they correspond to $d = 3.2$ ($H = 7.8$) and $d = 10.98$ ($H = 0.02$) km, respectively. It is seen that the estimated values of $\eta \cdot \tau_p(h_b, h_{end})$ are always lower than the values of the real optical





thickness, they are much lower in the cases of the 3D cloud (NUBF) compared to the plane-parallel cloud, and they strongly depend on the assigned value of $h_2$.

As in the case of the ground-based lidar, we can conclude that Eq. (5) does not take into account the escape effect. Therefore, it is only an approximate model of MS signals from space-based lidars.

## 4.2 Two-layered cloud

The two-layered cloud is the same as in Section 3.2, that is, two homogeneous layers are at the altitudes from 8 to 9 km and from 10 to 11 km. Every layer has the optical thickness $\tau_p = 1.0$; the total optical thickness is $\tau_p = 2.0$. Black and red points

in Fig. 9 show the results of our MC simulations reported in terms of the ratios $R_{MSto1}(d)$ and $R_{2to1}(d)$, respectively. The parameter $d$ is the distance measured from the cloud near edge, i.e. $d = 0.$ km corresponds to the altitude $H = 11.$ km. The cloud layers are within the ranges $d \in [0., 1.]$ and $d \in [2., 3.]$ km. The intervals $d \in ]1., 2.[$ and $d \in ]3., 11.]$ km (the altitudes $H$ from 10. to 9. km and from 8. to 0. km) are the cloud-free molecular atmosphere. Figures 9a and 9b correspond to the water and cirrus cloud, respectively.

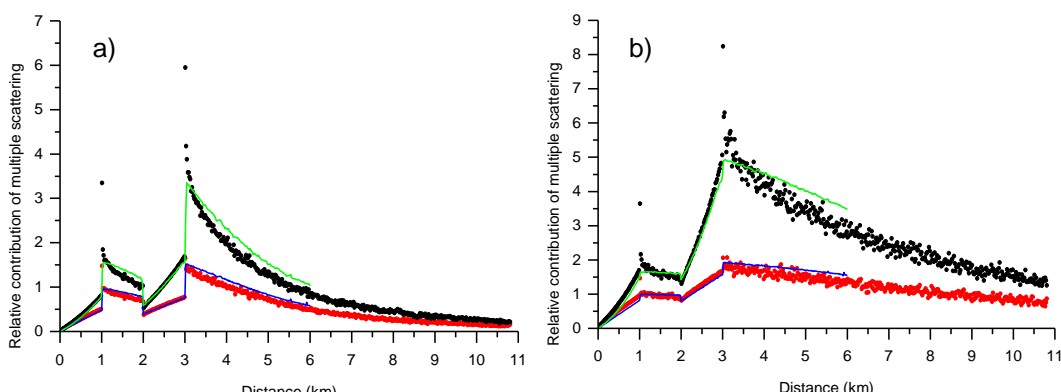


**Figure 9.** Monte Carlo simulations of multiple scattering $R_{MSto1}$ (black points) and double scattering $R_{2to1}$ (red points) relative contributions to lidar signals; (a) water cloud, (b) jet-stream cirrus. Eloranta-model simulations are green (MS) and blue (double scattering) curves.

The main properties of our MC simulations in Fig. 9 are in agreement with the results of the work (Flesia and Starkov, 1996); the distinctions are due to differences in the phase functions and the configuration characteristics. In addition, the vertical resolution in our work is 7.5 times better, and therefore the ratios $R_{MSto1}(d)$ and $R_{2to1}(d)$ are seen in more details.

The general behaviour of MC data around $d = 1.$ km and within the range $d \geq 3.$ km in Fig. 9 is the same as it is within the range $d \geq 3.$ km in Fig. 6. Thus, we focus attention only on the interval around $d = 2.$ km, that is, on the near edge of the





second cloud layer. The difference between the cases of the ground-based and the space-borne lidars is better seen from comparison of Figs. 5a and 9a, i.e. the water clouds. In the case of the space-borne lidar, the sampling volume is so large (due to the large receiver footprint) that the majority of forward scattered photons remain within it. This leads to the value $R_{MSto1}(d = 2.0\ km)$, which is almost the same as $R_{MSto1}(d = 1.0\ km)$ (see Fig. 9a). In contrast, the sampling volume, i.e. the footprint, is narrow in the case of the ground-based lidar. The majority of forward scattered photons escape it when they

are going through the cloud-free molecular atmosphere within the interval $d \in ]1.,2.[$ km. This leads to the value $R_{MSto1}(d = 2.0\ km)$ close to zero (see Fig. 5a). Generally, the same features are observed in Figs. 5b and 9b. They are less pronounced because the forward scattering pick of the phase function of cirrus cloud is much narrower.

     The relative contributions $R_{MSto1}(d)$ and $R_{2to1}(d)$ computed using the EM are shown in Fig. 9 by green and blue lines, respectively. The simulations were performed with the same values of the ratios $\mathcal{P}_n(\pi,h)/\mathcal{P}_1(\pi,h), n = 2, \dots, 5$ as in Section

4.1.1. Generally, the EM curves follow well the MC results. It is seen another time that the EM slightly overestimates the MS contribution in the cloud-free intervals and it is not able to reproduce the effect related to the pulse stretching.

## 5 Conclusions

     We performed Monte Carlo simulations of single-wavelength lidar signals from multi-layered clouds with special attention focused on features of multiple-scattering (MS) effect in regions of the cloud-free molecular atmosphere, i.e. between layers

or outside a cloud system. Despite the fact that the strength of lidar signals from molecular atmosphere is much lower compared to the in-cloud intervals, studies of MS effects in such regions are of interest from scientific and practical points of view. The results of this work are shown and discussed in terms of relative contributions of multiple scattering, that is, multiple-to-single scattering lidar-return ratios $R_{MSto1}(d)$. That provides possibility to accentuate the visibility of MS effects.

     The scattered photons emerging from a cloud do affect lidar signals received from the intervals of the cloud-free molecular

atmosphere. The MS effect is rather high within the region that is close to the far edge of a cloud. Those high values of the ratios $R_{MSto1}(d)$ are due to the features of the molecular backscattering, i.e. to the fact that the ratios $\mathcal{P}_n(\pi,h)/\mathcal{P}_1(\pi,h)$ are "equal to 1.0 due to the broad nature of the molecular phase function near the backscatter direction" (Whiteman et al., 2001; Eloranta, 1998). In the cases of the space-borne lidar, the additional MS contribution is due to the pulse stretching.

     The MS effect on lidar signals is decreasing with the increasing distance from the cloud far edge, i.e. the ratio $R_{MSto1}(d)$ tends

to zero. The decreasing is the direct consequence of the fact that the forward peak of particles phase functions is much larger than the receiver field of view. Therefore, the photons scattered within the forward peak escape the sampling volume formed by the RFOV, i.e. the escape effect. The escape effect is an inherent part of MS properties within the free atmosphere beyond the cloud far edge. That property is in direct contradiction with Eq. (5). Consequently, Eq. (5) is an approximate model of lidar signals under MS conditions.

The two-way transmittance method (see, e.g Young and Vaughan, 2009; Giannakaki et al. 2007) is based on Eq. (5) and used to deduce values of the apparent optical thickness of clouds. In view of the results of this work, it is advisable at least to choose



the reference point always at the same distance from the cloud far edge when estimating the apparent optical thickness of clouds.

In the cases of the ground-based lidar, the MS contribution is lower than 5% within the regions of the cloud-free molecular atmosphere having the distance from the cloud far edge about 1 km or higher. Therefore, it is safe to say that practitioners can use those regions as a reference and estimate the real optical thickness of clouds (see e.g. (Giannakaki et al. 2007) and references therein), if the EFOV and the RFOV are not very large. In the cases of the space-borne lidar, the decreasing rate of the MS contribution is so slow that the threshold of 5% can hardly be reached.

Using an example of a very simple 3D field, we demonstrated that the effect of non-uniform beam filling can be extremely strong in the case of a space-borne lidar. It is so strong, that, in our opinion, practitioners should employ with proper precautions lidar data from regions below the cloud base when treating data of a space-borne lidar. At the same time, it should be underlined that effects of the NUBF need further study, which will provide statistically significant results.

In the case of two-layered cloud, the distance of 1 km is sufficiently large that the scattered photons emerging from the first layer do not affect signals from the second layer when we are dealing with the ground-based lidar. In contrast, signals from the near edge of the second cloud layer are severely affected by the photons emerging from the first layer in the case of a space-borne lidar.

We evaluated the Eloranta model (EM) (Eloranta, 1998) in extreme conditions and showed its good performance in the cases of ground-based and space-borne (CALIOP) lidars. When the extinction coefficient is about $1.0$ km$^{-1}$ or lower, and the EFOV and the RFOV are quite narrow, five orders of scattering are sufficient to obtain satisfying accuracy of simulations. At the same time, we revealed the shortcoming that affects practical applications of the EM. Namely, values of the key parameters, i.e. of the ratios $\mathcal{P}_n(\pi, h)/\mathcal{P}_1(\pi, h)$ of phase functions in the backscatter direction for the $n^{th}$-order-scattered photon and a singly scattered photon depend not only on the particles phase function, but also on the distance from a lidar to the cloud and the receiver field of view. Values of the ratios $\mathcal{P}_n(\pi, h)/\mathcal{P}_1(\pi, h)$ vary within a quite large range. Therefore, the multiple scattering contribution to lidar signals can be largely overestimated or underestimated if erroneous values of the ratios are assigned to the EM. That problem can be circumvented by using Monte Carlo simulations or the empirical model (Shcherbakov et al., 2022) to calibrate the ratios $\mathcal{P}_n(\pi, h)/\mathcal{P}_1(\pi, h)$.

## Appendix A: Input parameters of the Eloranta model in homogeneous-cloud conditions

### A.1 Eloranta model

We have chosen to evaluate the Eloranta model (EM) (Eloranta, 1998) due to its following attractive features. The input parameters have clear physical meaning. The contribution of each $n^{th}$-order of scattering can be computed separately. The corresponding code can be developed without much difficulty because multiple integrals are the core of the EM. The good performance of the Eloranta model (EM) (Eloranta, 1998) in homogeneous-cloud conditions has already been reported in the literature (see e.g. Eloranta, 1998; Donovan and Van Lammeren, 2001). In our opinion, our results above demonstrate the





reliability of the EM in extreme conditions, i.e. when the extinction coefficient has undergone a stepwise jump. The objective

of this appendix is to reveal some significant features of the EM's input parameters that are related to the particles phase

function.

We developed two versions of codes based on the EM. The first version corresponds to Eq. (8) of (Eloranta, 1998). In order

to avoid ambiguity, we rewrite that equation in a way that all functions, including $\varepsilon(x)$, $\gamma(x)$, and $\Theta_s(x)$, are assigned in the

coordinate system where the lidar is at $h = 0$ and $h$ is the distance from the lidar

$$R_{n\,to\,1}(h) = \frac{\mathcal{P}_n(\pi,h)}{\mathcal{P}_1(\pi,h)}\left[1 - \exp\left(-\frac{\rho_t^2}{\rho_l^2}\right)\right]^{-1} \cdot \left\{ \int_{-d}^{d} \gamma(h - |x_1|) \cdot \varepsilon(h - |x_1|) \int_{x_1}^{d} \gamma(h - |x_2|) \cdot \varepsilon(h - |x_2|) \cdots \int_{x_{n-2}}^{d} \gamma(h - |x_{n-1}|) \cdot \right.$$

$$\varepsilon(h - |x_{n-1}|) \cdot \left[1.0 - \exp\left(-\frac{\rho_t^2 h^2}{x_1^2 \Theta_s^2(h-|x_1|) + x_2^2 \Theta_s^2(h-|x_2|) + \cdots + x_{n-1}^2 \Theta_s^2(h-|x_{n-1}|) + \rho_l^2 h^2}\right)\right] dx_{n-1} \cdots dx_2\, dx_1 \right\}. \qquad \text{(A1)}$$

$n \geq 2$, $\varepsilon(h) = \varepsilon_p(h) + \varepsilon_m(h)$, $\varepsilon_p(h)$ and $\varepsilon_m(h)$ are the particles and the molecular extinction coefficients, respectively. The

function $\gamma(h)$ is the fraction of the energy in the forward peak of the phase function, $h_b$ is the distance from the lidar to the

cloud near edge, $d = h - h_b$. If calculations are performed only within a cloud, $d$ is just the cloud penetration depth. As usual,

the notation $|x_i|$ means the absolute value of $x_i$. $\rho_t$ is the half-angle of the receiver field of view, $\rho_l$ is the half-angle of the

emitter divergence. $\Theta_s$ is the $1/e$ diffraction peak angular half width (Whiteman et al., 2001), in other words, $\Theta_s$ is the

parameter that characterises a Gaussian approximation for the forward scattered peak. $\Theta_s$ can be estimated using the

relationship (Eloranta, 1998; Hogan, 2006)

$$\Theta_s = \lambda/(\pi r_G), \qquad \text{(A2)}$$

where $r_G$ is the equivalent-area radius of the particles size distribution, $\lambda$ is the wavelength.

Other input parameters of the EM are the ratios $\mathcal{P}_n(\pi,h)/\mathcal{P}_1(\pi,h)$ of phase functions in the backscatter direction for the $n^{th}$-

order-scattered photon and a singly scattered photon (Whiteman et al., 2001). It is assumed that the backscattered phase

function $\mathcal{P}_n(\pi,h)$ for $n^{th}$-order scattering is independent of angle near 180° with a value, which is a weighted average near the

backscatter direction (Eloranta, 1998). The following is underlined in the work (Eloranta, 1998): "For observations it will be

necessary to use assumed values. For typical phase functions, $\mathcal{P}_n(\pi,h)/\mathcal{P}_1(\pi,h)$ is between 0.5 and 1."

The second version of the EM code correspond to Eq. (11) of (Eloranta, 1998), where the constant value $\gamma(x) = 1/2$ was

assumed using a reference to diffraction theory. We reproduce the equation for $n^{th}$-order of scattering with the reformulation

outlined in (Eloranta, 2000; Whiteman et al., 2001) (see Eq. (13) in Eloranta, 2000) and using our notations

$$R_{n\,to\,1}(h) = \frac{\mathcal{P}_n(\pi,h)}{\mathcal{P}_1(\pi,h)}\left[1 - \exp\left(-\frac{\rho_t^2}{\rho_l^2}\right)\right]^{-1} \cdot \left\{ \frac{\tau^{n-1}(h_b,h)}{(n-1)!} - \int_{h_b}^{h} \varepsilon(x_1) \int_{x_1}^{h} \varepsilon(x_2) \cdots \int_{x_{n-2}}^{h} \varepsilon(x_{n-1}) \cdot \right.$$

$$\exp\left(-\frac{\rho_t^2 h^2}{(h-x_1)^2 \Theta_s^2(x_1) + (h-x_2)^2 \Theta_s^2(x_2) + \cdots + (h-x_{n-1})^2 \Theta_s^2(x_{n-1}) + \rho_l^2 h^2}\right) dx_{n-1} \cdots dx_2\, dx_1 \right\} \qquad \text{(A3)}$$

where $\tau(h_b,h) = \int_{h_b}^{h} \varepsilon(x)dx$ and other notations have the same meaning as in Eq. (A1).

We used Eqs. (A1) and (A3) only in cases when $h \geq h_b$; we have verified that the first version of the EM code, i.e. Eq. (A1),

gives the same results as Eq. (A3) if the constant value $\gamma(x) = 1/2$ is assigned in Eq. (A1). We tested our codes of the EM





against the data available in the literature. All test were performed with $\gamma(x) = 1/2$. We obtained perfect agreement with Figs.

6-8 of the work (Eloranta, 1998) using $\mathcal{P}_n(\pi, h)/\mathcal{P}_1(\pi, h) = 0.75, n = 2, 3, 4$ and with Fig. 15(b) of the work (Donovan and Van Lammeren, 2001) using $\mathcal{P}_n(\pi, h)/\mathcal{P}_1(\pi, h) = 1.0, n = 2, 3, 4$.

**A.1 Eloranta model**

The results reported in Fig. A1 were obtained for the same configuration that was described above (see Section 3.1). That is, a ground-based lidar is at the altitude $H = 0$. km; the distance to the clouds base is 8 km. The full RFOV is 1.0 mrad; the full

EFOV is 0.14 mrad. The emitter wavelength $\lambda$ is 0.532 µm. The single-layer homogeneous cloud is within the altitude range $H \in [8., 11.]$ km, has the optical thickness $\tau_p = 3.0$, i.e. cloud particles have the extinction coefficient $\varepsilon_p(h) = 1.0$ km$^{-1}$.

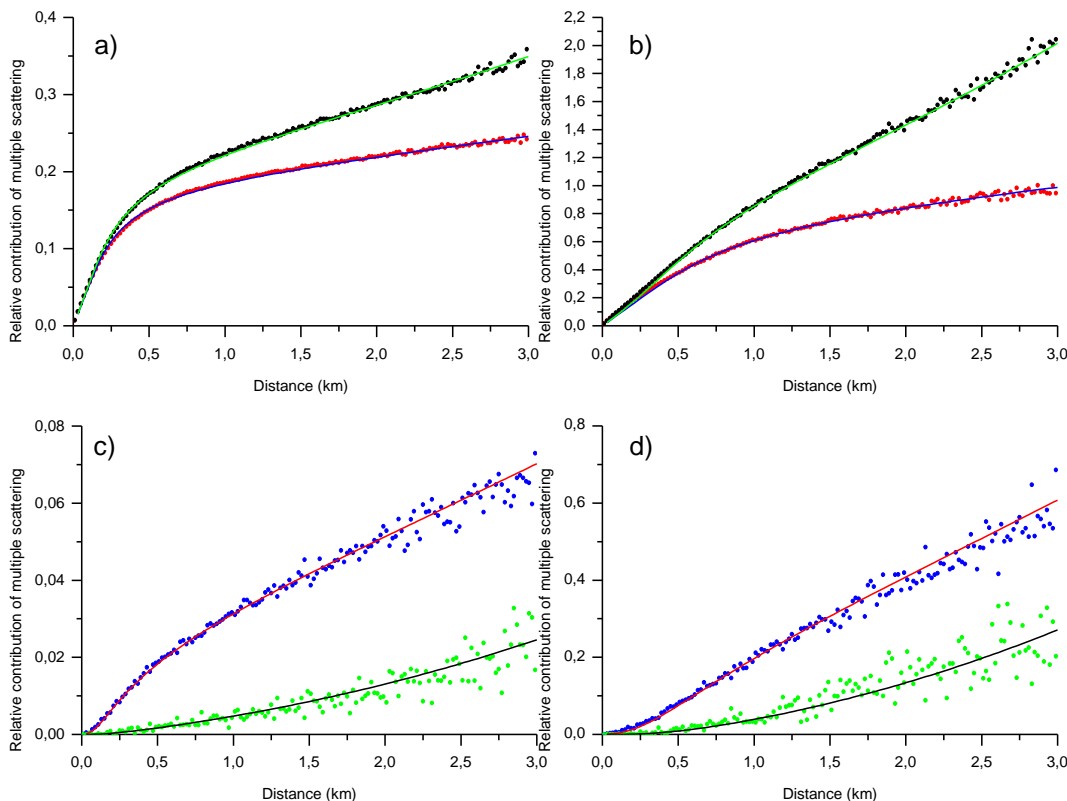

**Figure A1.** Monte Carlo simulations of multiple scattering $R_{MSto1}$ (black points), double scattering $R_{2to1}$ (red points), the third $R_{3to1}(d)$ (blue points) and the forth $R_{4to1}(d)$ (green points) orders, respectively, relative contributions to lidar signals;

(a, c) water cloud, (b, d) jet-stream cirrus. Eloranta-model simulations are green (MS), blue (double scattering), red (third order), and black (forth order) curves.





The results of the Monte-Carlo modeling (points) and the EM's simulations (lines) are shown in Fig. A1 only for the in-cloud range $d \in [0.,3.]$ km, i.e. the altitude interval $H \in [8., 11.]$ km. Figures A1a and A1c correspond to the water cloud, Figs. A1b

and A1d to cirrus cloud. The microphysical and optical parameters of cloud particles are done in Section 2.2 above. The results of each case are divided into two panels; Figs. A1a and A1c show the relative contribution of multiple $R_{MSto1}(d)$ (black points and green line) and double $R_{2to1}(d)$ (red points and blue line) scattering; Figs. A1b and A1d show the relative contribution of the third $R_{3to1}(d)$ (blue points and red line) and the forth $R_{4to1}(d)$ (green points and black line) orders. In other words, the lower panels are complementary to the corresponding upper panels. (Note that each panel in Fig. A1 has its own scale of the

y-axe.)

The good agreement between the MC and EM data is evident. That agreement was obtained by adjusting the EM parameters by the following way. The distance $h_b$ to the cloud near edge, the extinction coefficient $\varepsilon_p(d)$, the half-angle $\rho_t$ of the receiver field of view, and the half-angle $\rho_l$ of the emitter divergence were assigned according the configuration used for the MC simulations. First, we found the values of $\Theta_s$ and $\mathcal{P}_2(\pi)/\mathcal{P}_1(\pi)$ by fitting the MC data on $R_{2to1}(d)$ with the Eloranta model

using the ordinary least squares approach. Then we found the values of $\mathcal{P}_3(\pi)/\mathcal{P}_1(\pi)$ and $\mathcal{P}_4(\pi)/\mathcal{P}_1(\pi)$ by fitting the MC data on $R_{3to1}(d)$ and $R_{4to1}(d)$, respectively. We have limited our EM simulations by five orders of scattering. Consequently, the ratio $\mathcal{P}_5(\pi)/\mathcal{P}_1(\pi)$ was computed to fit the remained part of the total multiple scattering.

**Table A1.** Fitting values of the EM parameters.

|  | Ground based lidar | | Space based lidar | |
|---|---|---|---|---|
|  | Water cloud | JS cirrus | Water cloud | JS cirrus |
| $\Theta_s$ (rad) | 0.01882 | 0.0064 | 0.01882 | 0.0064 |
| $\Theta_s$ (degree) | 1.078 | 0.3667 | 1.078 | 0.3667 |
| $\mathcal{P}_2(\pi)/\mathcal{P}_1(\pi)$ | 0.502 | 0.7721 | 0.4993 | 0.8211 |
| $\mathcal{P}_3(\pi)/\mathcal{P}_1(\pi)$ | 0.395 | 0.66 | 0.5 | 0.99 |
| $\mathcal{P}_4(\pi)/\mathcal{P}_1(\pi)$ | 0.33 | 0.51 | 0.4 | 0.99 |
| $\mathcal{P}_5(\pi)/\mathcal{P}_1(\pi)$ | 0.26 | 0.553 | 0.35 | 0.99 |

The obtained values of parameters are shown in Table A1. The value of $\Theta_s$ is in good agreement with Eq. (A2) for the water cloud, it is about 7% higher for the jet-stream cirrus. The important result is the fact that the values of the ratios $\mathcal{P}_n(\pi)/\mathcal{P}_1(\pi)$, $n = 2, \ldots, 5$ are quite small, especially for the water cloud, and they decrease with the order of scattering increasing. If the recommendation of the work (Eloranta, 1998), i.e. "for typical phase functions, $\mathcal{P}_n(\pi, h)/\mathcal{P}_1(\pi, h)$ is between 0.5 and 1", is applied, the effect of the multiple scattering will be largely overestimated in the case of the water cloud. In our

opinion, the cause of small values of the ratios $\mathcal{P}_n(\pi)/\mathcal{P}_1(\pi)$ and of the declining $R_{MSto1}(d)$ in the free atmosphere range is common, i.e. photons, scattered in forward and backward directions, escape the sampling volume.





We performed the same kind of work for the case of space-borne lidar and the configuration described in Section 4.1.1. In other words, we provide the values of parameters when the same cloud was probed by the ground-based and the space-based lidars. The obtained values of parameters are shown in Table A1 as well. It is seen that the values of the ratios $\mathcal{P}_n(\pi)/\mathcal{P}_1(\pi)$

depend not only on the phase function of particles, but also on the lidar configuration, especially on the distance, the RFOV, and the EFOV.

*Data availability.* The data of simulations are available from the corresponding author upon request.

*Author contributions.* VS, FS, GM, and CC contributed to developing the McRALI code, numerical simulations, and data treatment. VS wrote the manuscript with help from FS. FS and GM acquired funding.

*Competing interests.* The contact author has declared that neither they nor their co-authors have any competing interests.

*Disclaimer.* Publisher's note: Copernicus Publications remains neutral with regard to jurisdictional claims in published maps and institutional affiliations.

*Acknowledgements.* This work is part of the French scientific community EECLAT project (Expecting EarthCARE, Learning from A-train). The EECLAT community and research activities are supported by the National Center for Space Studies (CNES)

and the National Institute for Earth Sciences and Astronomy (INSU).

*Financial support.* This research has been supported by the National Institute for Earth Sciences and Astronomy (INSU grant).

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
