# Peer review of "Multiple-scattering effects on single-wavelength lidar sounding of multi-layered clouds"

_Atmospheric Measurement Techniques, 2023_

## Author Comment (AC1)

Response to Reviewer # 1

We thank the reviewer for his review and valuable comments. The manuscript has been modified according to the suggestions proposed by the reviewer. The remainder is devoted to the specific response item-by-item of the reviewer's comments.

*RC=Reviewer Comments*
AR=Author response
TC=Text Changes

*General comments: The authors numerically simulated single-wavelength lidar multiple scattering signals by the Monte Carlo method, and focuses on the impacts of cloud multiple scattering on the lidar return signals in the cloud-free molecular atmosphere between cloud layers or outside a cloud layer. The author reported some interesting results, for example, the ratio of multiple scattering contributions to single scattering signals in the molecular atmosphere near the cloud edge is even larger than the ratio in the cloud (i.e., stepwise jump phenomena), and the multiple scattering effects are decreasing with the increase of the distance from the cloud edge (i.e., escape effects), and so on. It is worth publishing. However, there are several shortcomings in this paper.*

*First, why did the authors set the altitude of the cloud base to 8km? It is too high for water clouds. Why is the extinction coefficient of the water cloud set to 1.0 km^{-1}? It is too small. It is not representative for water clouds. The authors should give a reasonable explanation for this.*
We added to the revised manuscript (page 3 line 88) the following text.
The MS effect on lidar signals depends on a number of parameters, namely, the configuration (the distance to a cloud, the emitter field of view (EFOV), and the RFOV) as well as the cloud optical characteristics (the extinction coefficient, the albedo, the scattering matrix and so on) (see e.g. Shcherbakov et al., 2022; and references therein). Because the MC method is very time-consuming, it is not suited to take into account variations of all mentioned parameters. Therefore, our study was restricted to the cases when all cloud layers are within the range between the altitudes 8 and 11 km. Almost all MC simulations were performed for cloud particles having the extinction coefficient $\varepsilon_p(h) = 1.0$ km$^{-1}$ for the following reasons. On the one hand, technical capacities of contemporary lidars provide possibility to record signals from the cloud free atmosphere beyond the far edge of a cloud having the optical thickness $\tau_p = 3.0$. On the other hand, MS effect cannot be neglected and is clearly seen in a number of cases (Shcherbakov et al., 2022). Our choice of the parameters values was deliberate despite the fact that the altitude range $H \in [8., 11.]$ km does not correspond to the usual altitudes of warm clouds, the value $\varepsilon_p(h) = 1.0$ km$^{-1}$ is quite small for water clouds and rather high for cirrus clouds. With such a choice, the phase-function impact on multiple scattering is free of interference of other parameters variations.

*Secondly, the simulation results are particularly noisy. Is it convincing?*
We agree that our Monte-Carlo (MC) data are quite noisy, especially when we deal with the cases of the low extinction coefficient of the jet-stream cirrus probed by the ground-based lidar. At the same time, we believe that the quality of our MC simulations is sufficient to perform qualitative analysis of multiple scattering (MS) effects on lidar signals and justify our conclusions.
We recall that MC simulations are very time-consuming. For example, one case with $4 \cdot 10^{11}$ photons emitted by the lidar takes about 180 hours of the computing time ("DELL PowerEdge

R940 Server" with 20 jobs running in parallel). It would be preferable to reduce the random noise by 5 times in the cases of cirrus particles, but it would take about $180 \cdot 5^2 = 4500$ hours, which is not reasonable.

*Thirdly, there is logical confusion in the interpretation of the simulation results in Section 3.1.1, which is misleading. For example, the authors stated that the stepwise jump phenomenon is simply caused by the stepwise jump in the phase function at scattering angles close to 180 degrees. In my opinion, other factors (such as the molecular extinction coefficient) also might have effects on this.*

We have to underline that our analysis of the stepwise jump and the corresponding conclusions in Section 3.1.1 are first of all based on the properties of the ratio $R_{2to1}(d)$, i.e. the data of the MC simulations when no more than two scattering orders were taken into account. In other words, the ratio $R_{2to1}(d)$ corresponds to the ***Double Scattering*** (DS) approximation. We share the opinion that properties observed under the conditions of the DS approximation have to be analyzed using a corresponding approach (see details below).

We added to the revised manuscript (page 10 line 266) the following text.
It is of importance to underline that the discussion in this Section is mainly based on the data obtained under conditions of the double scattering (DS) approximation. Namely, the ratio $R_{2to1}(d)$ of the water cloud case is used as the base to explain the stepwise jump and the escape effect for the following reasons. The both effects are well pronounced in the data $R_{2to1}(d)$ in Fig. 2a, the DS accounts for more than 2/3 of the multiple scattering, and last but not least, the DS can be understood intuitively.

*The validation results in Fig 3a and 3b can only show that the free atmospheric signal is mainly affected by the forward scattering of clouds.*

It was demonstrated in the work Eloranta (1998) that one of the key parameters that govern MS effects on lidar signals is a weighted average of a phase function near the backscatter direction. The importance of that idea was underlined in relation to Raman lidar measurements (see Section 7.2 in Whiteman et al., 2001). We followed that idea and confirmed it using MC simulations. There is no the stepwise jump in $R_{2to1}(d)$ in Fig. 3b because the backscatter peak of the corresponding particles phase function $f_{Ch2}(\theta)$ is much larger compared to the case in Fig. 3a. (See details below.)

*The author mentioned pulse stretching many times and emphasized that their explanations are different from pulse stretching explanations. Can the author specifically point out the difference between the two?*

Our explanation is based on the idea developed in the work Eloranta (1998). That idea has nothing in common with the pulse stretching. Moreover, the Eloranta model (EM) ignores the pulse stretching (see below).
We reword the reasoning of Section 3.1.1 in more details and use as an example the ratio $R_{2to1}(d)$, i.e. the DS case, in Fig. 2a. The main question is why the values of $R_{2to1}(d)$ are so high immediately beyond the cloud far edge, even higher than within the cloud?
The first idea to explain that high values is the pulse stretching. i.e. some photons go through two scattering within the cloud, returns to the receiver and have the round-trip distance equal to the case of the single scattering from the range of the free atmosphere beyond the cloud far edge. The first scattering is within the volume of the cloud bounded by the EFOV, the second scattering is within the volume of the cloud bounded by the RFOV. Elementary geometrical reasoning, which takes into account the values of the EFOV, the RFOV, the distance to the cloud, and the geometrical thickness of the cloud, leads to the conclusion that the round-trip

distance of a double scattered photon can gain at the most 3.14 meters. Therefore, only the range $d \in ]3., 3.02]$ km of $R_{2to1}(d)$ can be somewhat affected by the double scattering within the cloud, i.e. by the pulse-length stretching. The interval $d > 3.02$ km remains unaffected. (We recall that our MC data were computed so that photons were integrated over the range gate 20 m, i.e. 0.02 km.)

The cases when photons go through two scattering within the clear molecular atmosphere cannot explain the high values of the ratio $R_{2to1}(d)$ because the molecular extinction coefficient is very low; $R_{MSto1} < 0.001$ for the ground-based lidar and the CALIOP configuration when the 1976 standard atmosphere (NOAA, 1976) is considered.

Consequently, the answer has to be found within the cases when one scattering (the first or the second) is within the cloud and another scattering (respectively, the second or the first) is within the clear molecular atmosphere. The explanation is based on the results of the work Eloranta (1998) where it was demonstrated that one of the key parameters that govern MS effects on lidar signals is a weighted average of a phase function near the backscatter direction (see Eq. (10) of Eloranta, (1998)). That parameter depends on the width of the backward peak of the phase function (other factors being the same). It is considered using as ***multiplying coefficients*** the ratios $\mathcal{P}_n(\pi, h)/\mathcal{P}_1(\pi, h)$ in the Eloranta model, where $n$ is the order of scattering. The ratio $\mathcal{P}_2(\pi, h)/\mathcal{P}_1(\pi, h) \approx 0.5$ for the phase function of the water cloud in Fig 2a and $\mathcal{P}_2(\pi, h)/\mathcal{P}_1(\pi, h) = 1.0$ for the Rayleigh scattering. Therefore, the values of $R_{2to1}(d)$ are lower within the cloud and higher immediately beyond the cloud far edge.

*Finally, this paper is rather lengthy, and the simulation description is repeated many times. It is suggested that the author refine the text to improve the reading experience.*

Each section of the manuscript deals with a peculiar configuration. At the same time, there are some aspects that are common to other sections. In order to avoid misunderstanding, we explicitly describe each configuration. We believe that such repetitions are unavoidable.

**Specific comments**

*Line 62-66 in Section 1: "results of MC simulations published in the literature evidenced the following. As it is expected, lidar signals from regions of the cloud-free molecular atmosphere … are affected by the scattered light emerging from clouds." Has the literature analyzed this multiple scattering effect in detail? If so, what is the difference between the results discussed by the authors and them? The authors should make a detailed discussion on this, which is the main contribution of this paper.*

We added to the revised manuscript (page 2 line 65) the following text.

To our knowledge, optical processes and cloud characteristics that govern the effect on lidar signals of the emerging light, i.e. the MS effect, as well as its distinctive features have not been addressed in details in the literature.

*Section 2.1: Subscript "MS" is used in several physical quantities and has different physical meanings, which are easy to cause misunderstanding. For example, it denotes total order-scattering in the lidar signal S_{MS}, while the contribution of single scattering is excluded in the ratio R_{MSto1}. In contrast, the ratio R'_{MS} contains the contribution of single scattering.*

The subscript "MS" means that the MS effect is taken into account (is not neglected). In our opinion, the parameter $R_{MSto1}$ is defined in the text without ambiguity.

*Section 2.1: I suggest providing a simple schematic diagram to describe the lidar sounding of the cloud-free atmosphere outside the clouds. It will be especially helpful for understanding formulas (especially Eq. (8-14)).*

We added to the manuscript the **Supplement**. Figure S1 of the Supplement and the corresponding text give examples of the lidar sounding of the cloud-free atmosphere outside the clouds. We added to the revised manuscript (page 10 line 275) the following text.
See Fig. S1 and the explanation in the Supplement.

*Line 92-95 in Section 2.2: As seen from Fig. 1, the phase function f_{ch2} is the same as the f_{ch1} except for the backscattering directions. Does this mean that f_{ch2} does not meet the normalization condition? If so, it would significantly increase the backscattering contribution of lidar signals, and the difference between Figs. 3a and 3b will be easily understood. See the comment "Line 265-293 in Section 3.1.1". The author should briefly elaborate on it.*
All phase functions of our work were normalized properly. Moreover, the McRALI software automatically normalizes all phase functions that are downloaded from the input data in order to prevent such kind of errors of unexperienced users.

*Line 209, 215, and 386-387: Why is the extinction coefficient of the water cloud set to 1.0 km^{-1}? It is too small. It is not representative for water clouds. Why did the authors set the altitude of the cloud bottom to 8km? It is too high for water clouds. The authors should give a reasonable explanation for this. Alternatively, the authors at least emphasize the scope of application of the conclusions made in this paper in the abstract or conclusion.*
We added to the revised manuscript (page 3 line 88) the corresponding text (see above).
In addition we have to underline that most of our results are presented so that they remain unaltered when the lidar pointing angle and/or the layer altitude vary provided that the distance to the cloud near edge remains unchanged. It is due to the fact that the distance to a layer is one of the key parameters, which govern the effect of MS on lidar signals.
For example, if a water cloud extends from the surface to the range 5.1 – 7.0 km above and a ground based lidar is tilted by 50.5 degrees with respect to the zenith, the curves of Fig. 2a can be used to assess MS effects.

*Fig. 2 in Section 3.1: The simulation results are particularly noisy in Fig. 2. Is it convincing? Can the authors reproduce the simulation results using other Monte Carlo programs?*
The stepwise jumps in Figs. 2a and 5a (the water clouds) are well pronounced, they are largely beyond the range of the statistical errors of our MC simulations. We deliberately do not smooth our data giving a reader the possibility to see the level of the MC statistical errors.
We added to the manuscript the following text
McRALI software was thoroughly tested against data available in the literature (see Appendix in Alkasem et al., 2017). We perfume tests when new data are published (under the condition that a paper provides all input data necessary to reproduce simulations). For example, we obtained very good agreement with Fig. 4 of the work by Wang et al. (2021) (including the linear and the circular polarization degree). The good agreement is with data for both ground-based and space-borne lidars.
We can add that we are going to perform profound comparison with MSCART code (Wang et al., 2021). Unfortunately, for this very moment, we are not able to overcome problems of software compatibility.

*Line 261-262 in Section 3.1.1: "It means that only the range d ∈]3., 3.02] km of R2to1(d) can be somewhat affected by the pulse-length stretching." Why is only the range in [3,3.02] affected by a little pulse-length stretching? According to Miller and Stephens, 1999, if multiple scattering (except the exact forward scattering with zero scattering angle) occurs, the half of light path length will be greater than the sounding ranges under the single scattering approximation, and pulse stretching will occur. From this point of view, the multiple scattering*

*signals in the cloud-free atmosphere can be considered as caused by pulse stretching, but the influence degree is different.*

The ratio $R_{2to1}(d)$ was computed under the conditions of the double scattering (DS) approximation. Elementary geometrical reasoning, which takes into account the values of the EFOV, the RFOV, the distance to the cloud, and the geometrical thickness of the cloud, leads to the conclusion that the round-trip distance of a double scattered photon can gain at the most 3.14 meters. Therefore, only the range $d \in ]3., 3.02]$ km of $R_{2to1}(d)$ can be somewhat affected by the double scattering within the cloud.

In our opinion, general ideas of multiple scattering are not suitable for discussing results obtained within the DS approximation.

We added to the manuscript the **Supplement**. Figure S1 of the Supplement and the corresponding text give examples of the lidar sounding of the cloud-free atmosphere outside the clouds. We added to the revised manuscript (page 10 line 275) the following text.

(See Fig. S1 and the explanation in the Supplement.) In addition, we have to underline the following. The EM uses the assumption that "the multiply scattered photons are scattered from the same slab as the single-scattered photons" (see p. 2466 in Eloranta (1998)). To put it differently, the EM ignores the pulse stretching. Nevertheless, it reproduces with good accuracy the stepwise jumps in Fig. 2 on the base of the phase functions parameters.

*Line 262-264 in Section 3.1.1: "Thus, it is safe to assume that the stepwise jump of $RMSto1(d)$ and $R2to1(d)$ is due to the stepwise jump in phase-function properties at angles close to 180° (the phase function of particles within the cloud and the Rayleigh scattering within the free atmosphere)." It is too vague to be misleading. Do the authors mean that the stepwise jump of $R\_\{MSto1\}$ is caused by the stepwise jump in the phase function of the water cloud within the scattering angles close to 180 degrees or the significant difference between the phase function of the water cloud and the Rayleigh scattering at scattering angles close to 180 degrees? I prefer to think of it as the latter. However, I think other factors (such as the molecular extinction coefficient) also have effects on this.*

See the answer below as well as the revisions of the manuscript text.

In addition we have to underline that the molecular extinction coefficient is so low that $R_{MSto1} < 0.001$ for the ground-based lidar and the CALIOP configuration when the 1976 standard atmosphere (NOAA, 1976) is considered.

*Line 265-293 in Section 3.1.1: "That assumption is confirmed by the plots in Figs. 3a and b". It's not convincing. It can only show that the phase function of the water cloud at scattering angles close to 180 degrees can affect the lidar multiple scattering signals in the cloud, but not in the cloud-free atmosphere. Therefore, it can show that the free atmospheric signal is mainly affected by the forward scattering of clouds. However, it is hard to demonstrate that the stepwise jump pattern occurring in the cloud-free atmosphere is due to the stepwise jump in phase-function properties at angles close to 180°.*

We demonstrated that (i) the stepwise jump in $R_{2to1}(d)$ is due to the distinctive property ($\mathcal{P}_2(\pi,h)/\mathcal{P}_1(\pi,h) = 1$) of the Rayleigh phase function, and (ii) the stepwise jump disappears if the cloud phase function has the same property. The stepwise jump disappeared because the values of $R_{2to1}(d)$ became higher for the in-cloud range.

We have never stated that the phase function of the water cloud at scattering angles close to 180 degrees can affect values of $R_{2to1}(d)$ in the cloud-free atmosphere.

In view of the reviewer's comments above, we revised the text of the corresponding paragraph as follows.

There is no a stepwise jump immediately beyond the cloud far edge. It means that the component $G_3(\pi - \theta)$ of $f_{Ch2}(\theta)$ is large enough to have the weighted average equal to $f_{Ch2}(\pi)$

as in the case of the Rayleigh phase function. If we use the terms of the Eloranta model (Eloranta, 1998), the ratio $\mathcal{P}_2(\pi, h)/\mathcal{P}_1(\pi, h) = 1$ for the phase function $f_{Ch2}(\theta)$, i.e. it has the same value as in the case of the Rayleigh phase function. To put it differently, a higher proportion of photons is scattered by the cloud in the backward direction within the RFOV and contribute to lidar signals in the MS case and under the DS approximation. As a consequence, the ratios $R_{MSto1}(d)$ and $R_{2to1}(d)$ of Fig. 3b are much higher than in Figs. 2a and 3a for the in-cloud range $d \in [0., 3.]$ km.

*In my opinion, it can be more clearly explained by the small-angle forward scattering theory, i.e., the lidar signal is contributed by the light that experiences a series of forward scattering from the emitter, then a single backward scattering, and finally a series of forward scattering back to the receiver. The forward scattering is roughly the same for both the cloud and the free atmosphere. They all occur in the cloud. The difference is the backscattering events, one occurring in the cloud and one occurring in the free atmosphere. Therefore, I claim that both the molecular extinction coefficients and phase function are the main reason for the stepwise jump in the R_{MSto1}.*

We have to underline another time that our reasoning is first of all based on the MC results obtained under the conditions of the double scattering approximation and the DS accounts for more than 2/3 of the multiple scattering in Fig. 2a. In our opinion, explanations based on general ideas of the small-angle forward scattering theory cannot be applied to the DS approximation.

*In addition, the author mentioned pulse stretching many times and emphasized that their explanations are different from pulse stretching explanations. Can the author specifically point out the difference between the two?*

Our explanation is based on the idea of the work Eloranta (1998), which states that one of the key parameters that govern MS effects on lidar signals is a weighted average of a phase function near the backscatter direction. See details above.

We added to the manuscript the **Supplement**. Figure S2 of the Supplement and the corresponding text give the intuitive explanation.

We added to the revised manuscript (page 10 line 275) the following text.

See Fig. S2 and the intuitive explanation in the Supplement.

*Line 294-302 in Section 3.1.1: The explanation of the "escape effect" is incomplete. In addition to the large forward diffraction ring relative to the field of view, the particularly small cloud extinction coefficient should also be a major reason. This is because if the extinction coefficient is large, the average free path length of light is short, and it is difficult to escape from the sampling volume even after many times of scattering. That's why I asked the authors to explain why they chose this special case.*

MS effect on lidar signals depends on a set of parameters (see Shcherbakov et al., (2022) and references therein); the extinction coefficient is one of them. The well-known equation for lidar signals under MS conditions (see Eq. (5) of the manuscript) takes into account the extinction coefficient, but it ignores the escape effect. In our opinion, the extinction coefficient is not the key parameter to explain the escape effect. At the same time, we can accept that some features of the escape effect could depend on the cloud extinction coefficient. That question needs profound study.

In Section 3.1, we demonstrated that the RFOV (see Figs. 5a and 5b) and the cloud phase function (see Fig. 2) belong to the set of key parameters. It follows from comparison of the results for the ground-based and space-borne lidars that the distance to the cloud is also a key parameter.

Due to the fact that we assigned the same distance and optical thickness to the water and cirrus cloud, we are able to demonstrate that the decreasing rate of the relative contribution of MS outside the cloud depends on the forward-peak width of the phase function.

*Line 309-314 in Section 3.1.1: "the pulse stretching is the cause of the jump" This judgment seems to be too subjective. Although the field of view is large, the extinction coefficient is too small and thus the free path length is large, so the light may not be able to experience enough scattering to produce obvious pulse stretching.*

Elementary geometrical reasoning, which takes into account the values of the EFOV, the RFOV=110 mrad, the distance to the cloud, and the geometrical thickness of the cloud, leads to the conclusion that the round-trip distance when a photon is double scattered within the cloud can gain at the most 311.5 meters. In Figs. 5c and 5d the ratio $R_{2to1}(d)$ clearly exhibits the distinctive feature within the range $d \in ]3., 3.25]$ km, which is agreement with that estimation, i.e. the suggestion of the pulse stretching.

We recall that the ratios $R_{MSto1}(d)$ and $R_{2to1}(d)$ are the ***relative contributions***; they represent data, which are ***normalized*** by the lidar signal obtained under single scattering approximation.

*Line 408-410 in Section 4.1.1: "Thus, the stepwise jump in those cases is not only due to the stepwise jump in phase-function properties for angles close to 180°. We can suggest that the range $d \in ]3., 3.1]$ km is somewhat affected by the pulse stretching." This judgment seems to be too subjective. Can the authors specifically show the difference between the two explanations? It is the same question in the comment "Line 265-293 in Section 3.1.1"*

Our suggestion follows from the fact that the EM model reproduces well the stepwise jumps in Fig. 2 (the ground-based lidar, i.e. small footprint and low impact of multiple scattering) and is not able to reproduce the amplitude of the stepwise jumps in $R_{MSto1}(d)$ in Fig. 6 (the CALIOP configuration, i.e. large footprint and high impact of multiple scattering). Knowing that the EM ignores the pulse stretching (see above), it is reasonable to suggest that the difference between the MC and EM data is due to the pulse stretching.

*Line 430 Section 4.1.2: "The NUBF effect is so high in such conditions that it has to be shown in terms of lidar signals." It's expected, I think, and it's easy to predict. Can the author provide the lidar single scattering signal results in the case of a three-dimensional cloud field? This may be more interesting.*

Results of statistical analysis of NUBF effects on lidar signals from the cloud free atmosphere beyond the far edge of a field of cirrus clouds are the subject of our future work. Monte-Carlo simulations (both single scattering and multiple scattering conditions) will be done for of a realistic three-dimensional cirrus cloud field like in our work (Alkasem et al., 2017).

*Technical corrections*
We are grateful to the reviewer for providing the technical corrections.

*Line 52 in Section 1: "Using some cases as examples, good performance of approximate models was underlined by their authors." It is too vague to provide any useful information.*
That and next sentences of the manuscript have to be considered unseparated. On the one hand, we agree that good performance of approximate models was shown in corresponding papers. On the other hand, we believe that the accuracy level and the applicability bounds of approximate models still need to be rigorously evaluated.

*Line 75 in Section 1: "because multiple integrals are in its core. Therefore, it is an easy matter to develop the corresponding code." It seems that there is no logical relationship between the two, so it is suggested to modify it clearly.*

We added to the revised manuscript (page 1 line 75) the following text.
A code to compute multiple integrals belongs to the domain of basic programming tasks. Therefore, it is an easy matter to develop a code corresponding to the Eloranta model.

*Line 156 in Section 2.2: "matrixes" can be corrected to "matrices".*
Corrected in the revised manuscript.

*Line 191 in Section 2.2: "The Gaussian component $G2(\theta)$ is large" need be corrected to "The width of …".*
Corrected in the revised manuscript.

*Line 261 in Section 3.1.1: "the distance from the lidar to particles of the cloud layer is quite low" It is "short" not "low".*
Corrected in the revised manuscript.

*Line 337 in Section 3.1.2: "when and $\varepsilon p \leq 0.2$ km-1." The word "and" may need to be deleted.*
Corrected in the revised manuscript.

References
Alkasem, A., Szczap, F., Cornet, C., Shcherbakov, V., Gour, Y., Jourdan, O., Labonnote, L. C., and Mioche, G.: Effects of cirrus heterogeneity on lidar CALIOP/CALIPSO data, J. Quant. Spectrosc. Ra., 202, 38–49, https://doi.org/10.1016/j.jqsrt.2017.07.005, 2017.

Eloranta, E.: Practical model for the calculation of multiply scattered lidar returns, Appl. Opt., 37, 2464–2472, https://doi.org/10.1364/AO.37.002464, 1998.

NOAA: U.S. Standard Atmosphere: 1976, National Oceanographic and Atmospheric Administration, Washington, DC, USA, NOAA-S/T 76-1562, ASIN: B000MMZKDS, 1976.

Wang, Z, Zhang, J., and Gao, H.: Impacts of laser beam divergence on lidar multiple scattering polarization returns from water clouds, J. Quant. Spec. Rad. Trans., 268, 107618, https://doi.org/10.1016/j.jqsrt.2021.107618, 2021.

Whiteman, D. N., Evans, K. D., Demoz, B., Starr, D. O'C., Eloranta, E. W., Tobin, D., Feltz, W., Jedlovec, G. J., Gutman, S. I., Schwemmer, G. K., Cadirola, M., Melfi, S. H., and Schmidlin, F. J.: Raman lidar measurements of water vapor and cirrus clouds during the passage of Hurricane Bonnie, J. Geophys. Res., 106, 5211–5225, https://doi.org/10.1029/2000JD900621, 2001.

---

## Author Comment (AC2)

Response to Reviewer # 2

We thank the reviewer for his review and valuable comments. The manuscript has been modified according to the suggestions proposed by the reviewer. The remainder is devoted to the specific response item-by-item of the reviewer's comments.

*RC=Reviewer Comments*
AR=Author response

*The detailed analyses are clear and important in this manuscript. The authors have taken novel piece of work, which is of great interest to focus on peculiarities of MS effect in regions of the cloud-free molecular atmosphere and to evaluate performance of an approximate model with the focus on cloud-free regions. For this reason, I felt and consider the subject of this paper is interesting. The results are relevant, and appropriate for the journal. The whole paper is straight forward, the conclusion is convincing but not well written. I believe the manuscript can be made publishable, but will require significant revisions.*

*Is there any limitation for the model used in this manuscript? In the instrumentation/method part, the uncertainties and precision of datasets obtained from LIDAR (ground mounted-space-borne) should be elaborately discussed. It is suggested to include the limitations, accuracy levels with some of its values (scattering characteristics of LIDAR signals, which leads to resolve the uncertainties) in tabular form.*
We added to the manuscript Appendix B, where we provide relative errors of Monte Carlo modeling in tabular form and discuss their properties.

*As I found some mismatch, so suitable tools/formats should be followed thoroughly e.g., math tool should be used to write parameters of equations, non-italic. In addition, different formats of braces and square brackets were found; it should be presented in a consistent way. Spelling errors (may be typo) should be omitted e.g. Mont-Carlo simulations or Monte-Carlo simulations (see section 2.2). Numerous different corrections and suggestion are made in PDF form so have a look on them and carry on accordingly.*
In the revised manuscript, all equations and parameters of equations are written using "MS Word 2013 Equation Editor"; the formats of braces and square brackets were automatically assigned by that tool.
The manuscript was proofread, spelling errors and typos were corrected.
Unfortunately, we have not succeeded to find the suggestion that were made in PDF form. Nevertheless, we are grateful to the reviewer for the care.

*It is just on a lighter note: are the authors in position to present extinction coefficient vertical profiles of LIDAR, if any, I urge to have certain studies that definitely make the studies worth more. The authors should use and cite updated research work, not before 2010. It should include more details on the specific algorithms or techniques used to differentiate between single and multiple-scattering events.*
We added to the manuscript Figure 6 that shows vertical profiles of signals in the case of the space-borne lidar. If the reviewer suggests to show the extinction coefficient vertical profiles retrieved from lidar data, we would like to underline that retrievals, i.e. solutions to an inverse problem, have to be a subject of a separate work because need extensive explanations of used retrieval technics.
We added the reference on the work by Wang et al. (2021) in order to underline that we obtained very good agreement with Fig. 4 of that work.
In our opinion, "specific algorithms or techniques used to differentiate between single and multiple-scattering events" are a subject for a review like (Bissonnette, 2005). The review has to well elaborated, it cannot be within a work devoted a specific aspect of multiple scattering.

*Is there any scope to present the statistics of interesting results in tabular form? If so, i urge to include in the revised version. I suggest the authors to devote adequate time to proof read the manuscript correcting typos and grammar. Also the level of language used could be improved to depict some scholarly writing.*

Our results are available in tabular form from the corresponding author upon request. The most important numerical data are given in the tables of the manuscript.

The manuscript was proofread, grammar errors and typos were corrected.